# Is RLHF More Difficult than Standard RL?
# A Theoretical Perspective

**Yuanhao Wang,   Qinghua Liu,   Chi Jin**
Princeton University
{yuanhao,qinghual,chij}@princeton.edu

## Abstract

Reinforcement learning from Human Feedback (RLHF) learns from *preference* signals, while standard Reinforcement Learning (RL) directly learns from *reward* signals. Preferences arguably contain less information than rewards, which makes preference-based RL seemingly more difficult. This paper theoretically proves that, for a wide range of preference models, we can solve preference-based RL directly using existing algorithms and techniques for reward-based RL, with small or no extra costs. Specifically, (1) for preferences that are drawn from reward-based probabilistic models, we reduce the problem to robust reward-based RL that can tolerate small errors in rewards; (2) for general arbitrary preferences where the objective is to find the von Neumann winner, we reduce the problem to multiagent reward-based RL which finds Nash equilibria for factored Markov games under a restricted set of policies. The latter case can be further reduced to adversarial MDP when preferences only depend on the final state. We instantiate all reward-based RL subroutines by concrete provable algorithms, and apply our theory to a large class of models including tabular MDPs and MDPs with generic function approximation. We further provide guarantees when K-wise comparisons are available.

## 1   Introduction

Reinforcement learning (RL) is a control-theoretic problem in which agents take a sequence of actions, receive *reward feedback* from the environment, and aim to find good policies that maximize the cumulative rewards. The reward labels can be objective measures of success (winning in a game of Go [Silver et al., 2017]) or more often hand-designed measures of progress (gaining gold in DOTA2 [Berner et al., 2019]). The empirical success of RL in various domains [Mnih et al., 2013, Vinyals et al., 2019, Todorov et al., 2012] crucially relies on the availability and quality of reward signals. However, this also presents a limitation for applying standard reinforcement learning when designing a good reward function is difficult.

An important approach that addresses this challenge is reinforcement learning from Human feedback (RLHF), where RL agents learn from *preference feedback* provided by humans. Preference feedback is arguably more intuitive to human users, more aligned with human values and easier to solicit in applications such as recommendation systems and image generation [Pereira et al., 2019, Lee et al., 2023]. Empirically, RLHF is a key ingredient empowering the successes in tasks ranging from robotics [Jain et al., 2013] to large language models [Ouyang et al., 2022].

A simple observation about preference feedback is that preferences can always be reconstructed from reward signals. In other words, preference feedback contains arguably less information than scalar rewards, which may render the RLHF problem more challenging. A natural question to ask is:

**Is preference-based RL more difficult than reward-based RL?**

37th Conference on Neural Information Processing Systems (NeurIPS 2023).

Existing research on preference-based RL [Chen et al., 2022, Pacchiano et al., 2021, Novoseller et al., 2020, Xu et al., 2020, Zhu et al., 2023] has established efficient guarantees for learning a near-optimal policy from preference feedback. These works typically develop specialized algorithms and analysis in a white-box fashion, instead of building on existing techniques in standard RL. This leaves it open whether it is necessary to develop new theoretical foundation for preference-based RL in parallel to standard reward-based RL.

This work presents a comprehensive set of results on provably efficient RLHF under a wide range of preference models. We develop new simple reduction-based approaches that solve preference-based RL by reducing it to existing frameworks in reward-based RL, with little to no additional cost:

- For utility-based preferences—those drawn from a reward-based probabilistic model (see Section 2.1), we prove a reduction from preference-based RL to reward-based RL with robustness guarantees via a new Preference-to-Reward Interface (P2R, Algorithm 1). Our approach incurs no sample complexity overhead and the human query complexity [1] does not scale with the sample complexity of the RL algorithm. We instantiate our framework for comparisons based on (1) immediate reward of the current state-action pair or (2) cumulative reward of the trajectory, and apply existing reward-based RL algorithms to directly find near-optimal policies for RLHF in a large class of models including tabular MDPs, linear MDPs, and MDPs with low Bellman-Eluder dimension, etc. We further provide complexity guarantees when K-wise comparisons are available.

- For general (arbitrary) preferences, we consider the objective of the *von Neumann winner* [see, e.g., Dudík et al., 2015, Kreweras, 1965], a solution concept that always exists and extends the Condorcet winner. We reduce this problem to multiagent reward-based RL which finds Nash equilibria for a special class of factored two-player Markov games under a restricted set of policies. When preferences only depend on the final state, we prove that such factored Markov games can be solved by both players running Adversarial Markov Decision Processes (AMDP) algorithms independently. For preferences that depend on entire trajectory, we develop an adapted version of optimistic Maximum Likelihood Estimation (OMLE) algorithm [Liu et al., 2022a], which handles this factored Markov games under general function approximation.

Notably, our algorithmic solutions are either reductions to standard reward-based RL problems or adaptations of existing algorithms (OMLE). This suggests that technically, preference feedback is not difficult to address given the existing knowledge of RL with reward feedback. Nevertheless, our impossibility results for utility-based preference (Lemma 2, 3) and reduction for general preferences also highlight several important conceptual differences between RLHF and standard RL.

## 1.1 Related work

**Dueling bandits.** Dueling bandits [Yue et al., 2012, Bengs et al., 2021] can be seen as a special case of preference based RL with $H = 1$. Many assumptions later applied to preference based RL, such as an underlying utility model with a link function [Yue and Joachims, 2009], the Plackett-Luce model [Saha and Gopalan, 2019], and the Condorcet winner [Zoghi et al., 2014] can be traced back to literature on dueling bandits. A reduction from preference feedback to reward feedback for bandits is proposed by Ailon et al. [2014]. The concept of the von Neumann winner, which we employ for general preferences, has been considered in Dudík et al. [2015] for contextual dueling bandits.

**RL from human feedback.** Using human preferences in RL has been studied for at least a decade [Jain et al., 2013, Busa-Fekete et al., 2014] and is later incorporated with Deep RL [Christiano et al., 2017]. It has found empirical success in robotics [Jain et al., 2013, Abramson et al., 2022, Ding et al., 2023], game playing [Ibarz et al., 2018] and fine-tuning large language models [Ziegler et al., 2019, Ouyang et al., 2022, Bai et al., 2022]. RL with other forms of human feedback, such as demonstrations and scalar ratings, has also been considered in previous research [Finn et al., 2016, Warnell et al., 2018, Arakawa et al., 2018] but falls beyond the scope of this paper.

**Theory of Preference-based RL.** For utility-based preferences, Novoseller et al. [2020], Pacchiano et al. [2021] and Zhan et al. [2023b] consider *tabular* or *linear* MDPs, and assume that the per-

---

[1]Throughout this paper, by sample complexity we mean the number of interactions with the MDP, while the query complexity refers the total number of calls of human evaluators.

trajectory reward is linear in trajectory features. Xu et al. [2020] also considers the tabular setting. However, instead of assuming an explicit link function, several structural properties of the preference is assumed which guarantee a Condorcet winner and ensure small regret of a black-box dueling bandit algorithm. Finally, recent work of Zhu et al. [2023], Zhan et al. [2023a] consider utility-based preferences in the offline setting, assuming the algorithm is provided with a pre-collected human preference (and transition) dataset with good coverage. Compared to the above works, this paper considers the online setting and derives results for utility-based preferences with general function approximation, which are significantly more general than those for tabular or linear MDPs.

For general preferences, Chen et al. [2022] also develops sample-efficient algorithms for finding the von Neumann winner [2]. Their algorithm is computationally inefficient in general even when restricted to the tabular setting. Compared to this result, our AMDP-based reduction algorithm (Section 4.2) is computationally efficient in the tabular or linear setting when the comparison only depends on the final states. For general trajectory-based comparison, our results apply to a richer class of RL problems including POMDPs.

Finally, we remark that all prior results develop specialized algorithms and analysis for preference-based RL in a white-box fashion. In contrast, we develop reduction-based algorithms which can directly utilize state-of-the-art results in reward-based RL for preference-based RL. This reduction approach enables the significant generality of the results in this paper compared to prior works.

## 2 Preliminaries

We consider reinforcement learning in episodic MDPs, specified by a tuple $(H, \mathcal{S}, \mathcal{A}, \mathbb{P})$. Here $\mathcal{S}$ is the state space, $\mathcal{A}$ is the action space, and $H$ is the length of each episode. $\mathbb{P}$ is the transition probability function; for each $h \in [H]$ and $s, a \in \mathcal{S} \times \mathcal{A}$, $\mathbb{P}_h(\cdot|s, a)$ specifies the distribution of the next state. A *trajectory* $\tau \in (\mathcal{S} \times \mathcal{A})^H$ is a sequence of interactions $(s_1, a_1, \cdots, s_H, a_H)$ with the MDP. A Markov policy $\pi = \{\pi_h : \mathcal{S} \to \Delta_{\mathcal{A}}\}_{h \in [H]}$ specifies an action distribution based on the current state, while a general policy $\pi = \{\pi_h : (\mathcal{S} \times \mathcal{A})^{h-1} \times \mathcal{S} \to \Delta_{\mathcal{A}}\}_{h \in [H]}$ can choose a random action based on the whole history up to timestep $h$.

In RLHF, an algorithm interacts with a reward-less MDP environment and may query comparison oracle (human evaluators) for preference information. We consider two types of preferences: utility-based ones and general ones.

### 2.1 Utility-based preferences

For utility-based comparison, we assume there exists an underlying reward function $r^\star : (\mathcal{S} \times \mathcal{A})^H \to [0, 1]$. Given a reward function, the value of a policy can be defined as $\mathbb{E}_\pi[\sum_{h=1}^H r^\star(s_h, a_h)]$, *i.e.* the expected cumulative reward obtained by executing the policy. An optimal policy is one that maximizes $\mathbb{E}_\pi[\sum_{h=1}^H r^\star(s_h, a_h)]$. We say that $\pi$ is $\epsilon$-optimal if $\mathbb{E}_\pi[\sum_{h=1}^H r^\star(s_h, a_h)] \geq \max_{\pi'} \mathbb{E}_{\pi'}[\sum_{h=1}^H r^\star(s_h, a_h)] - \epsilon$. We also consider a setting where the utility is only available for a whole trajectory. In this case, we assume that there is an underlying trajectory reward function $r^\star : (\mathcal{S} \times \mathcal{A})^H \to [0, H]$, which assigns a scalar utility to each trajectory. In this case the value of a policy can be defined similarly as $\mathbb{E}_{\tau \sim \pi}[r(\tau)]$.

In preference based RL, the reward is assumed to be unobservable, but is respected by the comparison oracle which models human evaluators.

**Definition 1** (Comparison oracle). *A comparison oracle takes in two trajectories $\tau_1$, $\tau_2$ and returns*

$$o \sim \mathrm{Ber}(\sigma(r^\star(\tau_1) - r^\star(\tau_2))),$$

*where $\sigma(\cdot)$ is a link function, and $r^\star(\cdot)$ is the underlying reward function.*

Here $\mathrm{Ber}(p)$ denotes a Bernoulli distribution with mean $p$. The comparison outcome $o = 1$ indicates $\tau_1 \succ \tau_2$, and vice versa. We additionally require that the inputs $\tau_1$, $\tau_2$ to the comparison oracle are *feasible* in the sense that they should be actual trajectories generated by the algorithm and cannot be synthesized artificially. This is motivated by the potential difficulty of asking human evaluators to compare out-of-distribution samples (e.g. random pixels).

---

[2] While their paper makes Assumption 3.1 in Chen et al. [2022] that seemingly assumes a Condorcet winner, this assumption actually always holds as $\pi^\star$ can be a general randomized policy, so their solution concept coincides with the von Neumann winner.

In this work, we also consider $(s, a)$ **preferences**, where $\Pr[(s_1, a_1) \succ (s_2, a_2)] = \sigma(r(s_1, a_1) - r(s_2, a_2))$. For notational simplicity, we will use $\tau$ to denote a state-action pair $(s, a)$ (which can be thought as an incomplete trajectory) so that $(s, a)$ preferences can be seen as a special case of the comparison oracle (Definition 1). See more details in Remark 1.

## 2.2 General preferences

For general preferences, we assume that for every trajectory pair $\tau, \tau'$, the probability that a human evaluator prefers $s$ over $s'$ is

$$M[\tau, \tau'] = \Pr[\tau \succ \tau']. \tag{1}$$

A general preference may not be realizable by a utility model, so we cannot define the optimal policy in the usual sense. Instead, we follow Dudík et al. [2015] and consider an alternative solution concept, the *von Neumann winner* (see Definition 5).

## 2.3 Function approximation

We first introduce the concept of eluder dimension, which has been widely used in RL to measure the difficulty of function approximation.

**Definition 2** (eluder dimension). *For any function class $\mathcal{F} \subseteq (\mathcal{X} \to \mathbb{R})$, its Eluder dimension* $\dim_{\mathrm{E}}(\mathcal{F}, \epsilon)$ *is defined as the length of the longest sequence* $\{x_1, x_2, \ldots, x_n\} \subseteq \mathcal{X}$ *such that there exists* $\epsilon' \geq \epsilon$ *so that for all* $i \in [n]$, $x_i$ *is* $\epsilon'$-independent *of its prefix sequence* $\{x_1, \ldots, x_{i-1}\}$, *in the sense that there exists some* $f_i, g_i \in \mathcal{F}$ *such that*

$$\sqrt{\sum_{j=1}^{i-1} \left( (f_i - g_i)(x_j) \right)^2} \leq \epsilon' \quad \text{and} \quad |(f_i - g_i)(x_i)| \geq \epsilon'.$$

Intuitively, eluder dimension measures the number of worst-case mistakes one has to make in order to identify an unknown function from the class $\mathcal{F}$. It is often used as a sufficient condition to prove sample efficiency guarantees for optimism-based algorithms.

As in many works on function approximation, we assume knowledge a class of reward functions $\mathcal{R}$ and realizability.

**Assumption 1** (Realizability). $r^\star \in \mathcal{R}$.

We further use $\overline{\mathcal{R}} := \{r + c | c \in [-H, 0], r \in \mathcal{R}\}$ to denote the reward function class augmented with a bias term. The inclusion of an additional bias is due to the assumption that preference feedback is based on reward *differences*, so we could only learn $r^\star$ up to a constant. We note that for the $d$-dimension linear reward class $\mathcal{R}_{\mathrm{linear}}$, $\dim_{\mathrm{E}}(\overline{\mathcal{R}}_{\mathrm{linear}}) \leq \widetilde{O}(d)$, and that for a general reward class $\mathcal{R}$, $\dim_{\mathrm{E}}(\overline{\mathcal{R}}, \epsilon) \leq \mathcal{O}(H \dim_{\mathrm{E}}(\mathcal{R}, \epsilon/2)^{1.5}/\epsilon)$. The details can be found in Appendix B.

# 3 Utility-based RLHF

Given access to comparison oracles instead of scalar rewards, a natural idea is to convert preferences back to scalar reward signals, so that standard RL algorithms can be trained on top of them. In this section, we introduce an efficient reduction from RLHF to standard RL through a preference-to-reward interface. On a high level, the interface provides approximate reward labels for standard RL training, and only queries the comparison oracle when uncertainty is large. The reduction incurs small sample complexity overhead, and the number of queries to human evaluators does not scale with the sample complexity of the RL algorithm using it. Moreover, it solicits feedback from the comparison oracle in a limited number of batches, simplifying the training schedule.

## 3.1 A preference-to-reward interface

The interaction protocol of an RL algorithm with the Preference-to-Reward (P2R) Interface is shown in Fig. 1. P2R maintains a confidence set of rewards $B_r$. When the RL algorithm wishes to learn the reward label of a trajectory $\tau$, P2R checks whether $B_r$ approximately agrees on the reward of $\tau$. If so, it can return a reward label with no queries to the comparison oracle; if not, it will query the comparison oracle on $\tau$ and a fixed trajectory $\tau_0$, and update the confidence set of reward functions. The details of P2R are presented in Algorithm 1.

---

**Algorithm 1** Preference-to-Reward (P2R) Interface

---

1: $\mathcal{B}_r \leftarrow \mathcal{R}, \mathcal{D} \leftarrow \{\}, \mathcal{D}_{\text{hist}} \leftarrow \{\}$
2: Execute the random policy to collect $\tau_0$
3: **Upon query of trajectory $\tau$:**
4: **if** $(\hat{r}, \tau) \in \mathcal{D}_{\text{hist}}$ **then**
5:     Return $\hat{r}$
6: **if** $\max_{r,r' \in \mathcal{B}_r} (r(\tau) - r(\tau_0)) - (r'(\tau) - r'(\tau_0)) < 2\epsilon_0$ **then**
7:     $\hat{r} \leftarrow r(\tau) - r(\tau_0)$ for an arbitrary $r \in \mathcal{B}_r$
8:     $\mathcal{D}_{\text{hist}} \leftarrow \mathcal{D}_{\text{hist}} \cup (\hat{r}, \tau)$
9: **else**
10:     Query comparison oracle $m$ times on $\tau$ and $\tau_0$; compute average comparison result $\bar{o}$
11:     $\hat{r} \leftarrow \arg\min_{x \in [-H,H]} |\sigma(x) - \bar{o}|, \quad \mathcal{D} \leftarrow \mathcal{D} \cup (\hat{r}, \tau), \quad \mathcal{D}_{\text{hist}} \leftarrow \mathcal{D}_{\text{hist}} \cup (\hat{r}, \tau)$
12:     Update $\mathcal{B}_r$
$$\mathcal{B}_r \leftarrow \left\{ r \in \mathcal{B}_r : \sum_{(\hat{r}, \tau) \in \mathcal{D}} \left( r(\tau) - r(\tau_0) - \hat{r} \right)^2 \leq \beta \right\}$$

13: Return $\hat{r}$

---

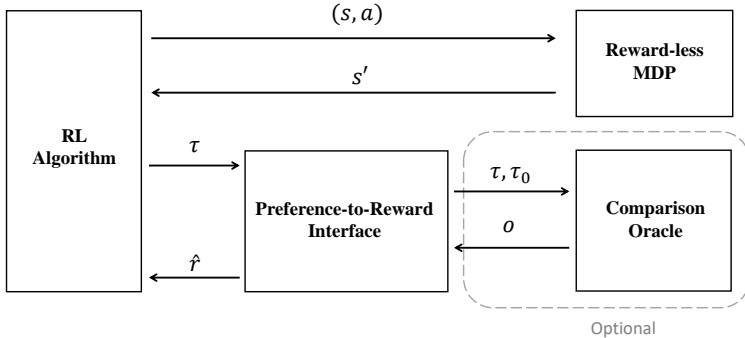

Figure 1: Interaction protocol with the reward learning interface.

The performance of P2R depends on the RL algorithm running on top of it. In particular, we define sample complexity of a standard reward-based RL algorithm $\mathscr{A}$ as follows.

**Definition 3.** *An RL algorithm $\mathscr{A}$ is $g(\epsilon)$-robust and has sample complexity $\mathcal{C}(\epsilon, \delta)$ if it can output an $\epsilon$-optimal policy using $\mathcal{C}(\epsilon, \delta)$ samples with probability at least $1 - \delta$, even if the reward of each trajectory $\tau$ is perturbed by $\varepsilon(\tau)$ with $\|\varepsilon(\tau)\|_\infty \leq g(\epsilon)$.*

We would like to note that the requirement of being $g(\epsilon)$-robust is typically not restrictive. In fact, any tabular RL algorithm with sample complexity would be $O(\epsilon/H)$-robust with the same sample complexity, while many algorithms with linear function approximation are $O(\epsilon/\text{poly}(d, H))$-robust [Jin et al., 2020b, Zanette et al., 2020] (again with the same sample complexity). We can further show that one can effortlessly convert any standard RL algorithms with sample complexity $\mathcal{C}$ into an $O(1/\mathcal{C})$-robust one by using the procedure described in Lemma B.3.

**Remark 1** (Trajectory vs. $(s, a)$ preferences)**.** So far, we presented the comparison oracle (Definition 1) and the P2R (Algorithm 1) using trajectory rewards. This would require the RL algorithm using P2R to be one that learns from once-per-trajectory scalar reward. To be compatible with standard RL algorithms, we can formally set $\tau$ as an "incomplete trajectory" $(s, a)$ in both Definition 1 and Algorithm 1. This would not change the theoretical results regarding the P2R reduction.

**Theoretical analysis of P2R**    We assume that $\sigma(\cdot)$ is known and satisfies the following regularity assumption.

**Assumption 2.** $\sigma(0) = \frac{1}{2}$; for $x \in [-H, H], \sigma'(x) \geq \alpha > 0$.

Assumption 2 is common in bandits literature [Li et al., 2017] and is satisfied by the popular Bradley-Terry model, where $\sigma$ is the logistic function. It is further motivated by Lemma 2 and Lemma 3: when $\sigma$ is unknown or has no gradient lower bounds, the optimal policy can be impossible to identify.

**Lemma 2.** *If $\sigma$ is unknown, even if there are only two possible candidates, the optimal policy is indeterminate.*

**Lemma 3.** *If $\sigma'(\cdot)$ is not lower bounded, for instance when $\sigma(x) = \frac{1}{2}(1 + \text{sign}(x))$, the optimal policy is indeterminate.*

P2R enjoys the following theoretical guarantee when we choose $\epsilon_0 = g(\epsilon)/2$, $\beta = \frac{\epsilon_0^2}{4}$, $d_{\overline{\mathcal{R}}} = \dim_{\text{E}}(\overline{\mathcal{R}}, \epsilon_0)$ and $m = \Theta\left(\frac{d_{\overline{\mathcal{R}}} \ln(d_{\overline{\mathcal{R}}}/\delta)}{\epsilon_0^2 \alpha^2}\right)$.

**Theorem 4.** *Suppose that Assumption 1 and 2 hold, and that $\mathscr{A}$ is an $g(\epsilon)$-robust RL algorithm with sample complexity $\mathcal{C}(\epsilon, \delta)$. By running $\mathscr{A}$ with the interface in Algorithm 1, we can learn an $\epsilon$-optimal policy using $\mathcal{C}(\epsilon, \delta)$ samples and $\widetilde{\mathcal{O}}\left(\frac{d_{\overline{\mathcal{R}}}^2}{\alpha^2 g(\epsilon)^2}\right)$ queries to the comparison oracle with probability $1 - 2\delta$.*

The full proof of Theorem 4 is deferred to Appendix B.3. The key idea of the analysis is to use the fact that $r^\star \in \mathcal{B}_r$ and the condition in Line 6 to show that the returned reward labels $\hat{r}$ are approximate accurate, and to use properties of eluder dimension to bound the number of samples for which the comparison oracle is called.

Theorem 4 shows that P2R is a provably efficient reduction: any standard RL algorithm $\mathcal{A}$ combined with P2R induces a provably efficient algorithm that learns an approximately optimal policy from preference feedback. The number of required interactions with the MDP environment is identical to that of the standard RL algorithm, and the query complexity only scales with the complexity of learning the reward function.

**Comparison with other approaches**   A more straightforward reduction would be extensively querying the comparison oracle for every sample generated by the RL algorithm. While this direct reduction would not suffer from increased sample complexity, it encounters two other drawbacks: (1) the oracle complexity, or the total workload for human evaluators, increases proportionally with the sample complexity of the RL algorithm, which can be prohibitive; (2) the RL training would need to pause and wait for human feedback at every iteration, creating substantial scheduling overhead.

Another method to construct reward feedback is to learn a full reward function directly before running the RL algorithm, as in Ouyang et al. [2022]. However, without pre-existing high-quality offline datasets, collecting the samples for reward learning would require solving an exploration problem at least as hard as RL itself [Jin et al., 2020a], resulting in significant sample complexity overhead. In P2R, the exploration problem is solved by the RL algorithm using it.

Compared to the two alternative approaches, our reduction achieves the best of both worlds by avoiding sample complexity overhead with a query complexity that does not scale with the sample complexity.

## 3.2   Instantiations of P2R

When combined with existing sample complexity results, Theorem 4 directly implies concrete sample and query complexity bounds for preference-based RL in many settings, with no statistical and small computational overhead.

$(s, a)$ **preferences**   We first consider the comparison that is based on the immediate reward of the current state-action pair. Here we give tabular MDPs and MDPs with low Bellman-Eluder dimension [Jin et al., 2021] as two examples.

**Example 1** (Tabular MDPs). Our first example is tabular MDPs whose state and action spaces are finite and small. In this case $d_{\overline{\mathcal{R}}} = \widetilde{\mathcal{O}}(|\mathcal{S}||\mathcal{A}|)$. The UCBVI-BF algorithm, proposed by Azar et al. [2017], is a model-based tabular RL algorithm which uses upper-confidence bound value iteration with Bernstein-Freedman bonuses. UCBVI-BF has sample complexity $\mathcal{C}(\epsilon, \delta) = \mathcal{O}\left(H^3 |\mathcal{S}||\mathcal{A}|/\epsilon^2\right)$ and is $\mathcal{O}(\epsilon/H)$ robust due to Lemma B.4.

**Proposition 5.** *Algorithm 1 with UCBVI-BF learns an $\epsilon$-optimal policy of a tabular MDP from preference feedback using $\widetilde{\mathcal{O}}\left(\frac{H^3 |\mathcal{S}||\mathcal{A}|}{\epsilon^2}\right)$ episodes of interaction with the environment and $\widetilde{\mathcal{O}}\left(\frac{H^2 |\mathcal{S}|^2 |\mathcal{A}|^2}{\alpha^2 \epsilon^2}\right)$ queries to the comparison oracle. The algorithm is computationally efficient.*

**Example 2** (Low Bellman-eluder dimension). Bellman-eluder dimension [Jin et al., 2021] is a complexity measure for function approximation RL with a Q-value function class $\mathcal{F}$. It can be

shown that a large class of RL problems, including tabular MDPs, linear MDPs, reactive POMDPs and low Bellman rank problems, all have small Bellman-eluder dimension. Furthermore, Jin et al. [2021] designed an algorithm, named GOLF, which (i) first constructs a confidence set for the optimal Q-functions by including all the candidates with small temporal difference loss, (ii) then optimistically picks Q-estimate from the confidence set and executes its greedy policy, and (iii) repeats (i) and (ii) using the newly collected data. Assuming that $\mathcal{F}$ satisfies realizability and completeness property, GOLF is $g(\epsilon) = \Theta\left(\frac{\epsilon}{\sqrt{d_{\mathrm{BE}}}H^2}\right)$-robust with sample complexity $\mathcal{C}(\epsilon, \delta) = \widetilde{\mathcal{O}}\left(\frac{d_{\mathrm{BE}}H^4 \ln|\mathcal{F}|}{\epsilon^2}\right)$ where $d_{\mathrm{BE}}$ is the Bellman-eluder dimension of the problem. By applying Theorem 4, we immediately have the following result.

**Proposition 6.** *Algorithm 1 with GOLF [Jin et al., 2021] learns an $\epsilon$-optimal policy of RL problems with Bellman-Eluder dimension $d_{\mathrm{BE}}$ in $\widetilde{\mathcal{O}}\left(\frac{d_{\mathrm{BE}}H^4 \ln|\mathcal{F}|}{\epsilon^2}\right)$ episodes of interaction with the environment and $\widetilde{\mathcal{O}}\left(\frac{d_{\mathrm{BE}}d_{\mathcal{R}}^2 H^2}{\alpha^2\epsilon^2}\right)$ queries to the comparison oracle.*

**Trajectory-based preferences**  When the reward function is trajectory-based, we can instantiate P2R with the OMLE algorithm [Liu et al., 2022a] to solve any model-based RLHF of low generalized eluder dimension. In brief, OMLE is an optimism-based algorithm that maintains a model confidence set. This set comprises model candidates that demonstrate high log-likelihood on previously collected data. And in each iteration, the algorithm chooses a model estimate optimistically and executes its greedy policy to collect new data.

**Example 3** (Low generalized eluder dimension). Generalized eluder dimension [Liu et al., 2022a] is a complexity measure for function approximation RL with a transition function class $\mathcal{P}$. In Appendix D.1, we show that a simple adaptation of OMLE is $g(\epsilon) = \Theta(\frac{\epsilon}{\sqrt{d_{\mathcal{R}}}})$-robust with sample complexity $\mathcal{C}(\epsilon, \delta) = \widetilde{\mathcal{O}}\left(\frac{H^2 d_{\mathcal{P}}|\Pi_{\exp}|^2 \ln|\mathcal{P}|}{\epsilon^2} + \frac{Hd_{\mathcal{R}}|\Pi_{\exp}|}{\epsilon}\right)$, where $d_{\mathcal{P}}$ denotes the generalized eluder dimension of $\mathcal{P}$, $|\Pi_{\exp}|$ is a parameter in the OMLE algorithm, and $d_{\mathcal{R}} = \dim_{\mathrm{E}}(\mathcal{R}, \epsilon)$. Plugging it back into Theorem 4, we obtain the following result.

**Proposition 7.** *Algorithm 1 with OMLE learns an $\epsilon$-optimal policy of RL problems with low generalized eluder dimension in $\widetilde{\mathcal{O}}\left(\frac{H^2 d_{\mathcal{P}}|\Pi_{\exp}|^2 \ln|\mathcal{P}|}{\epsilon^2} + \frac{Hd_{\mathcal{R}}|\Pi_{\exp}|}{\epsilon}\right)$ episodes of interaction with the environment and $\widetilde{\mathcal{O}}\left(\frac{d_{\mathcal{R}}d_{\mathcal{R}}^2}{\alpha^2\epsilon^2}\right)$ queries to the comparison oracle.*

Liu et al. [2022a] prove that a wide range of model-based reinforcement learning problems have a low generalized eluder dimension $d_{\mathcal{P}}$ and only require a mild $|\Pi_{\exp}|$ to run the OMLE algorithm. Examples of such problems include tabular MDPs, factored MDPs, observable POMDPs, and decodable POMDPs. For a formal definition of generalized eluder dimension and more details on the aforementioned bound and examples, we refer interested readers to Appendix D.1 or Liu et al. [2022a]. Finally, we remark that it is possible to apply the P2R framework in other settings with different complexity measures, such as DEC [Foster et al., 2021] or GEC [Zhong et al., 2022], by making some minor modifications to the corresponding algorithms to ensure robustness.

## 3.3   Extension to $K$-wise comparison

In this subsection, we briefly discuss how our results can be extended to $K$-wise comparison assuming the following Plackett-Luce (PL) model [Luce, 1959, Plackett, 1975] of $K$ item preferences.

**Definition 4** (Plackett-Luce model). *The oracle takes in $K$ trajectories $\tau_1, \ldots, \tau_K$ and outputs a permutation $\phi : [K] \to [K]$ with probability $\mathbb{P}(\phi) = \prod_{k=1}^{K} \frac{\exp(\eta \cdot r(\tau_{\phi^{-1}(k)}))}{\sum_{t=k}^{K} \exp(\eta \cdot r(\tau_{\phi^{-1}(t)}))}$.*

Note that when $K = 2$, the above PL model reduces to a pair-wise trajectory-type comparison oracle (Definition 1) with $\sigma(x) = \exp(\eta x)/(\exp(\eta x) + 1)$ which satisfies Assumption 2 with $\alpha = \Theta(\eta \exp(-\eta H))$. The PL model satisfies the following useful property, which is a corollary of its internal consistency [Hunter, 2004].

**Property 8** (Hunter [2004, p396]). *For any disjoint sets $\{i_1, j_1\}, \ldots, \{i_k, j_k\} \subseteq [K]$, the following pair-wise comparisons are mutually independent: $\mathbf{1}(\phi(i_1) > \phi(j_1)), \ldots, \mathbf{1}(\phi(i_k) > \phi(j_k))$ where $\phi$ is a permutation sampled from $PL(\tau_1, \ldots, \tau_K)$. Moreover, $\mathbf{1}(\phi(i_m) < \phi(j_m)) \sim \mathrm{Ber}(\sigma(r(\tau_{i_m}) - r(\tau_{j_m})))$, where $\sigma(x) = \exp(\eta x)/(\exp(\eta x) + 1)$.*

This property enables "batch querying" the preferences on $\lfloor K/2 \rfloor$ pairs of $(\tau_1, \tau_2)$ in parallel, which returns $\lfloor K/2 \rfloor$ independent pairwise comparisons outcomes. This would enable us to reduce the number of queries by a factor of $\Omega(K)$ for small $K$ in both Algorithm 1 and 3.

**Theorem 9** (P2R with $K$-wise comparison). *Suppose $\mathscr{A}$ is an $g(\epsilon)$-robust RL algorithm with sample complexity $\mathcal{C}(\epsilon, \delta)$, and assume the same conditions and the same choice of parameters as in Theorem 4. By running $\mathscr{A}$ with the interface in Algorithm 1, we can learn an $\epsilon$-optimal policy using $\mathcal{C}(\epsilon, \delta)$ samples and $\widetilde{\mathcal{O}} \left( \frac{d_{\overline{\mathcal{R}}}^2}{\alpha^2 g(\epsilon)^2 \min\{K, m\}} \right)$ queries to the $K$-wise comparison oracle with probability $1 - 2\delta$.*

Theorem 9 is a direct consequence of Theorem 4: If $K \geq 2m$, we can obtain $m$ independent comparisons between two trajectories by a single query to the $K$-wise comparison oracle and therefore reduce the overall query complexity in Theorem 4 by a factor of $m$; otherwise, we can get $m$ independent comparisons by making $\mathcal{O}(m/K)$ queries to the $K$-wise comparison oracle, which reduces the overall query complexity by a factor of $K$.

However, the above parallelization benefits of using $K$-wise comparison might be an artifact of the PL model: it seems improbable that the same human evaluator would independently rank $\lfloor K/2 \rfloor$ copies of item $A$ and item $B$. It remains an interesting problem to develop $K$-wise comparison models more suitable to RLHF.

# 4 RLHF From General Preferences

The utility-based approach imposes strong assumptions on human preferences. Not only is the matrix $M[\tau, \tau']$ in (1) assumed to be exactly realizable by $\sigma(r(\tau) - r(\tau'))$, but also $\sigma$ is assumed to be known and have a gradient lower bound. Moreover, the utility-based approach assumes that *transitivity*: if $\Pr[\tau_1 \succ \tau_2] \geq 0.5$, $\Pr[\tau_2 \succ \tau_3] \geq 0.5$, then $\Pr[\tau_1 \succ \tau_3] \geq 0.5$. However, experiments have shown that human preferences can be intransitive [Tversky, 1969]. These limitations of the utility-based approach motivates us to consider general preferences.

A general preference may not be realizable by a utility model, so we cannot define the optimal policy in the usual sense. Instead, we follow Dudík et al. [2015] and consider an alternative solution concept, the *von Neumann winner*.

**Definition 5.** $\pi^\star$ *is the von Neumann winner policy if* $(\pi^\star, \pi^\star)$ *is a symmetric Nash equilibrium of the constant-sum game:* $\max_\pi \min_{\pi'} \mathbb{E}_{\tau \sim \pi, \tau' \sim \pi'} M[\tau, \tau']$.

The duality gap of the game is defined as

$$\mathrm{DGap}(\pi_1, \pi_2) := \max_\pi \mathbb{E}_{\tau \sim \pi, \tau' \sim \pi_2} M[\tau, \tau'] - \min_\pi \mathbb{E}_{\tau \sim \pi_1, \tau' \sim \pi} M[\tau, \tau'].$$

We say that $\pi$ is an $\epsilon$-approximate von Neumann winner if the duality gap of $(\pi, \pi)$ is at most $\epsilon$. The von Neumann winner has been studied under the name of *maximal lotteries* in the context of social choice theory [Kreweras, 1965, Fishburn, 1984]. The von Neumann winner is a natural generalization of the optimal policy concept for non-utility based preferences. It is known that

- Intuitively, the von Neumann winner $\pi^\star$ is a randomized policy that "beats" any other policy $\pi'$ in the sense that $\mathbb{E}_{\tau \sim \pi^\star, \tau' \sim \pi'} M[\tau, \tau'] \geq 1/2$;

- If the utility-based preference model holds and the transitions are deterministic, the von Neumann winner is the optimal policy;

- The von Neumann winner is the only solution concept that satisfies population-consistency and composition-consistency in social choice theory [Brandl et al., 2016].

Finding the von Neumann winner seems prima facie quite different from standard RL tasks. However, in this section we will show how finding the von Neumann winner can be reduced to finding the restricted Nash equilibrium in a type of Markov games. For preferences based on the final state, we can further reduce the problem to RL in adversarial MDP.

## 4.1 A reduction to Markov games

**Factorized and independent Markov games (FI-MG).** Consider a two-player zero-sum Markov games with state space $\mathcal{S} = \mathcal{S}^{(1)} \times \mathcal{S}^{(2)}$, action space $\mathcal{A}^{(1)}$ and $\mathcal{A}^{(2)}$ for each player respectively,

transition kernel $\{\mathbb{P}_h\}_{h\in[H]}$ and reward function $r$. We say the Markov game is factorized and independent if the transition kernel is factorized:

$$\mathbb{P}_h(s_{h+1}\mid s_h,a_h)=\mathbb{P}_h(s_{h+1}^{(1)}\mid s_h^{(1)},a_h^{(1)})\times\mathbb{P}_h(s_{h+1}^{(2)}\mid s_h^{(2)},a_h^{(2)}),$$

where $s_h=(s_h^{(1)},s_h^{(2)})$, $s_{h+1}=(s_{h+1}^{(1)},s_{h+1}^{(2)})$, $a_h=(a_h^{(1)},a_h^{(2)})\in\mathcal{A}^{(1)}\times\mathcal{A}^{(2)}$.

The above definition implies that the Markov game can be partitioned into two MDPs, where the transition dynamics are controlled separately by each player, and are completely independent of each other. The only source of correlation between the two MDPs arises from the reward function, which is permitted to depend on the joint trajectory from both MDPs. Building on the above factorization structure, we define *partial trajectory* $\tau_{i,h}:=(s_1^{(i)},a_1^{(i)},\ldots,s_h^{(i)})$ that consists of states of the $i$-th MDP factor and actions of the $i$-th player. Furthermore, we define a *restricted policy class* $\Pi_i$ that contains all policies that map a partial trajectory to a distribution in $\Delta_{\mathcal{A}_i}$, i.e.,

$$\Pi_i:=\Big\{\{\pi_h\}_{h\in[H]}:\ \pi_h\in\big((\mathcal{S}^{(i)}\times\mathcal{A}_i)^{h-1}\times\mathcal{S}^{(i)}\to\Delta_{\mathcal{A}_i}\big)\Big\}\ \text{for } i\in[2].$$

And the goal is to learn a restricted Nash equilibrium $(\mu^\star,\nu^\star)\in\Pi_1\times\Pi_2$ such that

$$\mu^\star\in\arg\max_{\mu\in\Pi_1}\mathbb{E}_{\tau\sim\mu,\tau'\sim\nu^\star}[r(\tau,\tau')]\quad\text{and}\quad\nu^\star\in\arg\min_{\nu\in\Pi_2}\mathbb{E}_{\tau\sim\mu^\star,\tau'\sim\nu}[r(\tau,\tau')].$$

**Finding von Neumann winner via learning restricted Nash.** We claim that finding an approximate von Neumann winner can be reduced to learning an approximate restricted Nash equilibrium in a FI-MG. The reduction is straightforward: we simply create a Markov game that consists of two independent copies of the original MDP and control the dynamics in the $i$-th copy by the $i$-th player's actions. Such construction is clearly factorized and independent. Moreover, the restricted policy class $\Pi_i$ is equivalent to the universal policy class in the original MDP. We further define the reward function as $r(\tau,\tau')=M[\tau,\tau']$ where $M$ is the general preference function. By definition 5, we immediately obtain the following equivalence relation.

**Proposition 10.** *If $(\mu^\star,\nu^\star)$ is a restricted Nash equilibrium of the above FI-MG, then both $\mu^\star$ and $\nu^\star$ are von Neumann winner in the original problem.*

The problem we are faced with now is how to learn restricted Nash equilibria in FI-MG. In the following sections, we present two approaches that leverage existing RL algorithms to solve this problem: (i) when the preference function depends solely on the final states of the two input trajectories, each player can independently execute an adversarial MDP algorithm; (ii) for general preference functions, a straightforward adaptation of the OMLE algorithm is sufficient under certain eluder-type conditions.

## 4.2 Learning from final-state-based preferences via adversarial MDPs

In this section, we consider a special case where the preference depends solely on the final states of the two input trajectories, i.e., $M(\tau,\tau')=M(s_H,s_H')$. Given the previous equivalence relation between von Neumann winner and restricted Nash in FI-MG, one natural idea is to apply no-regret learning algorithms, as it is well-known that running two copies of no-regret online learning algorithms against each other can be used to compute Nash equilibria in zero-sum normal-form games. Since this paper focuses on sequential decision making, we need no-regret learning algorithms for adversarial MDPs, which we define below.

**Adversarial MDPs.** In the adversarial MDP problem, the algorithm interacts with a series of MDPs with the same unknown transition but adversarially chosen rewards for each episode. Formally, there exists an unknown groundtruth transition function $\mathbb{P}=\{\mathbb{P}_h\}_{h=1}^H$. At the beginning of the $k$-th episode, the algorithm chooses a policy $\pi^k$ and then the adversary picks a reward function $r^k=\{r_h^k\}_{h=1}^H$. After that, the algorithm observes a trajectory $\tau^k=(s_1^k,a_1^k,y_1^k,\ldots,s_H^k,a_H^k,y_H^k)$ sampled from executing policy $\pi^k$ in the MDP parameterized by $\mathbb{P}$ and $r^k$, where $\mathbb{E}[y_h^k\mid s_h^k,a_h^k]=r_h^k(s_h^k,a_h^k)$. We define the *regret* of an adversarial MDP algorithm $\mathscr{A}$ to be the gap between the algorithm's expected payoff and the best payoff achievable by the best fixed Markov policy:

$$\text{Regret}_K(\mathscr{A}):=\max_{\pi\in\Pi_{\text{Markov}}}\sum_{k=1}^K\mathbb{E}_\pi\Big[\sum_{h=1}^H r_h^k(s_h,a_h)\Big]-\sum_{k=1}^K\mathbb{E}_{\pi_k}\Big[\sum_{h=1}^H r_h^k(s_h,a_h)\Big].$$

Now we explain how to learn a von Neumann winner via running adversarial MDP algorithms. We simply create two copies of the original MDP and instantiate two adversarial MDP algorithms $\mathscr{A}_1$ and

$\mathscr{A}_2$ to control each of them separately. To execute $\mathscr{A}_1$ and $\mathscr{A}_2$, we need to provide reward feedback to them in each $k$-th episode. Denote by $s_H^{k,(1)}$ and $s_H^{k,(2)}$ the final states $\mathscr{A}_1$ and $\mathscr{A}_2$ observe in the $k$-th episode. We will feed $y^k \sim \mathrm{Ber}(M(s_H^{k,(1)}, s_H^{k,(2)}))$ into $\mathscr{A}_1$ and $1 - y^k$ into $\mathscr{A}_2$ as their reward at step $H-1$, respectively. And all other steps have zero reward feedback. The formal pseudocode is provided in Algorithm 4 (Appendix E). The following theorem states that as long as the invoked adversarial MDP algorithm has sublinear regret, the above scheme learns an approximate von Neumann winner in a sample-efficient manner.

**Theorem 11.** *Suppose* $\mathrm{Regret}_K(\mathscr{A}) \leq \beta K^{1-c}$ *with probability at least* $1 - \delta$ *for some* $c \in (0, 1)$. *Then Algorithm 4 with* $K = (4\beta/\epsilon)^{1/c}$ *outputs an* $\epsilon$-*approximate von Neumann winner with probability at least* $1 - 2\delta$.

In order to demonstrate the applicability of Theorem 11, we offer two examples where sublinear regret can be achieved in adversarial MDPs via computationally efficient algorithms. The first one is adversarial tabular MDPs where the number of states and actions are finite, i.e., $|\mathcal{S}|, |\mathcal{A}| \leq +\infty$.

**Example 4** (adversarial tabular MDPs). Jin et al. [2019] proposed an algorithm $\mathscr{A}$ with $\mathrm{Regret}_K(\mathscr{A}) \leq \widetilde{\mathcal{O}}(\sqrt{|\mathcal{S}|^2|\mathcal{A}|H^3K})$. Plugging it into Theorem 11 leads to $K = \widetilde{\mathcal{O}}(|\mathcal{S}|^2|\mathcal{A}|H^3/\epsilon^2)$ sample complexity and query complexity for learning $\epsilon$-approximate von Neumann winner.

The second example is adversarial linear MDPs where the number of states and actions can be infinitely large while the transition and reward functions admit special linear structure. See Sherman et al. [2023] for the precise formulation of the adversarial linear MDP problem.

**Example 5** (adversarial linear MDPs). Sherman et al. [2023] proposed an algorithm $\mathscr{A}$ with $\mathrm{Regret}_K(\mathscr{A}) \leq \widetilde{\mathcal{O}}(dH^2K^{6/7})$ for online learning in adversarial linear MDPs.[3] Combining it with Theorem 11 leads to $K = \widetilde{\mathcal{O}}(d^7H^{14}/\epsilon^7)$ sample complexity and query complexity for learning $\epsilon$-approximate restricted Nash equilibria.

### 4.3 Learning from trajectory-based preferences via OMLE_equilibrium

In this section, we consider the more general case where the preference $M[\tau, \tau']$ is allowed to depend arbitrarily on the two input trajectories. Similar to the utility-based setting, we assume that the learner is provided with a preference class $\mathcal{M} \subseteq ((\mathcal{S} \times \mathcal{A})^H \times (\mathcal{S} \times \mathcal{A})^H \to [0, 1])$ and transition function class $\mathcal{P}$ a priori, which contains the groundtruth preference and transition we are interacting with. Previously, we have established the reduction from learning the von Neumann winner to learning restricted Nash in FI-MG. In addition, learning restricted Nash in FI-MG is in fact a special case of learning Nash equilibrium in partially observable Markov games (POMGs). As a result, we can directly adapt the existing OMLE algorithm for learning Nash in POMGs [Liu et al., 2022b] to our setting, with only minor modifications required to learn the von Neumann winner. We defer the algorithmic details for this approach (Algorithm 5) to Appendix F, and present only the theoretical guarantee here.

**Theorem 12.** *Suppose Assumption 3 holds. There exist absolute constant* $c_1$ *and* $c_2$ *such that for any* $(T, \delta) \in \mathbb{N} \times (0, 1]$ *if we choose* $\beta_{\mathcal{M}} = c_1 \ln(|\mathcal{M}|T/\delta)$ *and* $\beta_{\mathcal{P}} = c_1 \ln(|\mathcal{M}|T/\delta)$ *in Algorithm 5, then with probability at least* $1 - \delta$, *the duality gap of the output policy of Algorithm 5 is at most*

$$\frac{4\xi(d_{\mathcal{P}}, T, c_2\beta_{\mathcal{P}}, |\Pi_{\exp}|)}{T} + c_2\sqrt{\frac{d_{\mathcal{M}}\beta_{\mathcal{M}}}{T}},$$

*where* $d_{\mathcal{M}} = \dim_{\mathrm{E}}(\mathcal{M}, 1/T)$.

It has been proven that a wide range of RL problems admit a regret formulation of $\xi = \widetilde{\mathcal{O}}(\sqrt{d_{\mathcal{P}}\beta|\Pi_{\exp}|T})$ with mild $d_{\mathcal{P}}$ and $|\Pi_{\exp}|$ [Liu et al., 2022a]. These problems include, but are not limited to, tabular MDPs, factored MDPs, linear kernel MDPs, observable POMDPs, and decodable POMDPs. For more details, please refer to Appendix D.1 or Liu et al. [2022a]. For problems that satisfy $\xi = \widetilde{\mathcal{O}}(\sqrt{d_{\mathcal{P}}\beta_{\mathcal{P}}|\Pi_{\exp}|T})$, Theorem D.1 implies a sample complexity of

$$\widetilde{\mathcal{O}}\left(\frac{d_{\mathcal{P}}|\Pi_{\exp}|\ln|\mathcal{P}|}{\epsilon^2} + \frac{d_{\mathcal{M}} \cdot \ln|\mathcal{M}|}{\epsilon^2}\right).$$

The sample complexity for specific tractable problems can be derived by plugging their precise formulation of $\xi$ (provided in Appendix D.1) into the above bound.

---

[3]Sherman et al. [2023] requires the adversarial reward function to be linear in the feature mapping of linear MDPs. In Appendix E, we show the reward signal we constructed in Algorithm 4 satisfies such requirement.

## Acknowledgement

This work was partial supported by National Science Foundation Grant NSF-IIS-2107304 and Office of Naval Research Grant N00014-22-1-2253.

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

## A  Proofs of Impossibility Results

*Proof of Lemma 2.* Consider two link functions $\sigma_1(x) := \frac{1}{1+\exp(-x)}$ and

$$\sigma_2(x) := \frac{1}{2} + \alpha_1 x + \alpha_2(x - \alpha_3) \cdot I[|x| > \alpha_3],$$

where $\alpha_1 = 1, \alpha_2 = -0.484, \alpha_3 = 0.3$ Consider an MDP with $H = 2$ and initial state $s_0$. Suppose that there are three terminal states $s_1, s_2, s_3$, where we observe that the trajectory preferences only depend on the terminal state in the following way:

$$\Pr[s_1 \succ s_2] = 0.7, \Pr[s_2 \succ s_3] = 0.8, \Pr[s_1 \succ s_3] = 0.903.$$

This can be explained by both $\sigma_1$ with $r^{(1)} = \{s_0 : 0, s_1 : 0.847, s_2 : 0, s_3 : -1.386\}$, and $\sigma_2$ with $r^{(2)} = \{s_0 : 0, s_1 : 0.2, s_2 : 0, s_3 : -0.3\}$.

Suppose that state $s_0$ has two actions $a$ and $b$, which leads to distributions $\{0.39 : s_1, 0.61 : s_3\}$ and $\{1 : s_2\}$ respectively. Under $r^{(1)}$ and $r^{(2)}$, the optimal action would be $a$ and $b$ respectively. Therefore without knowledge of the link function, it is impossible to identify the optimal policy even with perfect knowledge of the transition and comparison probabilities. □

*Proof of Lemma 3.* Similarly, consider an MDP with $H = 2$ and initial state $s_0$. Suppose that there are three terminal states $s_1, s_2, s_3$, where we observe that

$$\Pr[s_1 \succ s_2] = 1, \Pr[s_2 \succ s_3] = 1, \Pr[s_1 \succ s_3] = 1.$$

Meanwhile the state $s_0$ has two actions $a$ and $b$, leading to distributions $\{0.5 : s_1, 0.5 : s_3\}$ and $\{1 : s_2\}$ respectively. In this case, both $r^{(1)} = \{s_0 : 0, s_1 : 0.5, s_2 : 0, s_3 : -1\}$ and $r^{(2)} = \{s_0 : 0, s_1 : 1, s_2 : 0, s_3 : -0.5\}$ fits the comparison results perfectly. However under $r^{(1)}$, the optimal action is $b$ while under $r^{(2)}$, the optimal action is $a$. □

## B  Proofs for P2R

### B.1  Properties of eluder dimension

We begin by proving two properties of the eluder dimension of $\overline{\mathcal{R}}$.

**Lemma B.1.** *Consider* $\mathcal{R}_{\text{linear}} := \{\theta^\top x\}$, *where* $x \in \mathcal{X} \subseteq \mathbb{R}^d$, $\|\theta\|_2 \leq \gamma$. *Then there exists absolute constant* $C$ *such that*

$$\dim_E(\overline{\mathcal{R}}_{\text{linear}}, \epsilon) \leq C(d + 1) \ln\left(1 + \frac{\gamma + H}{\epsilon}\right) + 1.$$

*Proof.* Note that

$$\theta^\top x + c = [\theta, c]^\top [x, 1].$$

Therefore $\overline{\mathcal{R}}_{\text{linear}}$ can be seen as a $(d + 1)$-dimensional linear function class with parameter norm $\|[\theta, c]\|_2 \leq \gamma + H$. The statement is then a direct corollary of Russo and Van Roy [2013, Proposition 6]. □

**Lemma B.2.** *For a general function class* $\mathcal{R}$ *with domain* $\mathcal{X}$ *and range* $[0, H]$, *for* $\epsilon < 1$,

$$\dim_E(\overline{\mathcal{R}}, \epsilon) \leq \mathcal{O}(H \dim_E(\mathcal{R}, \epsilon/2)^{1.5}/\epsilon).$$

*Proof.* Suppose that there exists a sequence

$$\{r_i, c_i, x_i, y_i\}_{i \in [m]},$$

where $r_i \in \mathcal{R}$, $c_i \in [-H, 0]$, $x_i \in \mathcal{X}$ and $y_i \in \mathbb{R}$, such that for all $i \in [m]$

$$|r_i(x_i) + c_i - y_i| \geq \epsilon, \quad \sum_{j<i} |r_i(x_j) + c_i - y_j|^2 \leq \epsilon^2.$$

By definition, $\dim_E(\overline{\mathcal{R}}, \epsilon)$ is the largest $m$ so that such a sequence exists. By the pigeon-hole principle, there exists a subset $I \subseteq [m]$ of size at least $k = \frac{m}{\lceil H/\epsilon_0 \rceil}$ and $\bar{c} \in [0, H - \epsilon_0]$ such that $\forall i \in I$, $c_i \in [\bar{c}, \bar{c} + \epsilon_0]$. Denote the subsequence indexed by $I$ as $\{r_i', c_i', x_i', y_i'\}_{i \in [k]}$. Define $\widetilde{y}_i := y_i' - \bar{c}$. Now consider the sequence $\{r_i', x_i', \widetilde{y}_i\}_{i \in [k]}$. By definition $\forall i \in [k]$

$$|r_i'(x_i') - \widetilde{y}_i| \geq \epsilon - \epsilon_0.$$

It follows that

$$\sum_{j<i} |r_i'(x_j) - \widetilde{y}_j|^2 = \sum_{j<i} |r_i'(x_j) - (y_j' - c_j')|^2 + 2\sum_{j<i} (\bar{c} - c_j')\left(r_i'(x_j) - (y_j' - c_j')\right) + \sum_{j<i}(\bar{c} - c_j')^2$$

$$\leq \epsilon^2 + 2\sqrt{k}\epsilon_0\epsilon + k\epsilon_0^2.$$

We can choose $\epsilon_0 := \left(\frac{H\epsilon^2}{16m}\right)^{1/3}$ so that $\epsilon_0 \leq \epsilon/(4\sqrt{k})$. Then we can guarantee

$$|r_i'(x_i) - \widetilde{y}_i| \geq 0.5\epsilon, \quad \sum_{j<i} |r_i'(x_j) - \widetilde{y}_j|^2 \leq 2\epsilon^2.$$

By Jin et al. [2021, Proposition 43],

$$k \leq \left(1 + \frac{2\epsilon^2}{(0.5\epsilon)^2}\right) \dim_E(\mathcal{F}, 0.5\epsilon).$$

In other words,

$$\frac{m}{\lceil H/\epsilon_0 \rceil} \leq 9\dim_E(\mathcal{R}, 0.5\epsilon),$$

which gives

$$\dim_E(\overline{\mathcal{R}}, \epsilon) \leq \frac{216H}{\epsilon} \cdot \dim_E(\mathcal{R}, 0.5\epsilon)^{1.5}.$$

$\square$

## B.2 Properties related to robustness

**Lemma B.3** (Robustness to perturbation). *Any RL algorithm $\mathscr{A}$ with sample complexity $\mathcal{C}(\epsilon, \delta)$ can be converted to an algorithm $\mathscr{A}'$ that is $\frac{1}{\mathcal{C}(\epsilon,\delta)}$-robust with sample complexity $\mathcal{C}(\epsilon, \delta/3)$.*

*Proof.* Consider the following modification of $\mathscr{A}$: instead of using reward $r$ directly, we project $r$ to $+2H$ and $-2H$ unbiasedly; that is, the algorithm receives the binarized rewards

$$b(r) := \left\{2H : \frac{1}{2} + \frac{r}{4H}, -2H : \frac{1}{2} - \frac{r}{4H}\right\}.$$

By the definition of sample complexity, when using samples of $b(r^\star)$, $\mathscr{A}$ outputs a policy $\pi_0$ for $r^\star$ that is $\epsilon$-optimal with probability $1 - \delta$ in $K := \mathcal{C}(\epsilon, \delta)$ episodes. Denote the trajectories generated by running $\mathscr{A}$ on $b(r^\star)$ by $\tau_1, \cdots, \tau_K$. Now suppose that for $\tau_k$, the reward label is perturbed from $b(r^\star(\tau_k))$ to $b(r_k')$ with $|r_k' - r^\star(\tau_k)| \leq \epsilon' := (\mathcal{C}(\epsilon, \delta))^{-1}$; denote the output policy of $\mathscr{A}$ by $\pi'$. It can be shown that

$$\left|\ln\left(\frac{b(r^\star(\tau_k)) = 2H}{b(r_k') = 2H}\right)\right| \leq \ln\left(1 + \frac{\epsilon'}{H}\right).$$

Therefore the total density ratio

$$\sup_{\vec{r} \in \{2H, -2H\}^K} \left|\sum_{k=1}^{K} \ln\left(\frac{b(r^\star(\tau_1)), \cdots, b(r^\star(\tau_K)) = \vec{r}}{b(r_1'), \cdots, b(r_K') = \vec{r}}\right)\right| \leq K \ln\left(1 + \frac{\epsilon'}{H}\right) \leq \frac{K\epsilon'}{H} \leq 1.$$

It follows that the density ratio between $\pi$ and $\hat{\pi}$ is also bounded by $e$. Therefore the probability that $\hat{\pi}$ is not $\epsilon$-optimal is at most $3\delta$. Rescaling $\delta$ proves the lemma. $\square$

**Lemma B.4.** *Any tabular RL algorithm $\mathscr{A}$ with sample complexity $\mathcal{C}(\epsilon, \delta)$ is $\epsilon/(4H)$ robust with sample complexity $\mathcal{C}(\epsilon/2, \delta)$.*

*Proof.* Suppose that $\mathscr{A}$ is run on perturbed rewards, where rewards for trajectory $\tau$ is changed by $\varepsilon(\tau)$. By definition, using $\mathcal{C}(\epsilon/2, \delta)$ samples and with probability $1 - \delta$, it outputs an $\epsilon/2$-optimal policy $\hat{\pi}$ with respect to the perturbed reward function $r + \varepsilon$, where $\|\varepsilon\|_\infty \leq \epsilon/4H$. Denote the value function of policy $\pi$ with respect to reward $r$ by $V^{\pi,r}$. Further denote the optimal value function with respect to $r$ by $\pi^\star$. It holds that for any policy $\pi$,

$$\left| V^{\pi,r} - V^{\pi+\varepsilon,r} \right| \leq \epsilon/4.$$

Therefore

$$V^{\pi^\star,r} - V^{\hat{\pi},r} \leq \epsilon/2 + V^{\pi^\star,r+\varepsilon} - V^{\hat{\pi},r+\varepsilon}$$
$$\leq \epsilon/2 + \epsilon/2 = \epsilon.$$

In other words $\hat{\pi}$ is indeed an $\epsilon$-optimal policy with respect to the unperturbed rewards $r$. $\qquad\square$

### B.3 Proof of Theorem 4

**Lemma B.5.** *With* $m = \Theta\left(\frac{\ln(1/\delta')}{\alpha^2 \epsilon'^2}\right)$, *for each* $\tau$ *such that the comparison oracle is queried, with probability* $1 - \delta'$,

$$|\hat{r}(\tau) - (r^\star(\tau) - r^\star(\tau_0))| \leq \epsilon'.$$

*Proof.* Suppose that the comparison oracle is queried for $\tau$ and the average outcome is $\bar{o}$. By Hoeffding bound, with probability $1 - \delta'$,

$$|\bar{o} - \sigma(r^\star(\tau) - r^\star(\tau_0))| \leq \sqrt{\frac{\ln(2/\delta')}{m}}.$$

Since $\hat{r}(\tau) = \operatorname{argmin}_{x \in [-H,H]} |\sigma(x) - \bar{o}|$,

$$|\sigma(\hat{r}(\tau)) - \bar{o}| \leq |\sigma(r^\star(\tau) - r^\star(\tau_0)) - \bar{o}| \leq \sqrt{\frac{\ln(2/\delta')}{m}}.$$

It follows that

$$|\sigma(\hat{r}(\tau)) - \sigma(r^\star(\tau) - r^\star(\tau_0))| \leq 2\sqrt{\frac{\ln(2/\delta')}{m}}.$$

By Assumption 2,

$$|\hat{r}(\tau) - (r^\star(\tau) - r^\star(\tau_0))| \leq \frac{1}{\alpha} \cdot |\sigma(\hat{r}(\tau)) - \sigma(r^\star(\tau) - r^\star(\tau_0))| \leq \frac{2}{\alpha}\sqrt{\frac{\ln(2/\delta')}{m}} \leq \epsilon'.$$

$\qquad\square$

**Lemma B.6.** *Set* $m = \Theta(\frac{d \ln(d/\delta)}{\epsilon_0^2 \alpha^2})$ *and* $\beta = \frac{\epsilon_0^2}{4}$. *With probability* $1 - \delta$, *the number of samples on which the comparison oracle is queried is at most* $\dim_E(\overline{\mathcal{R}}, \epsilon_0)$.

*Proof.* Define $\tilde{r}^\star(\tau) := r^\star(\tau) - r^\star(\tau_0)$, $\tilde{r}(\tau) := r(\tau) - r(\tau_0)$.

When the comparison oracle is queried, $\max_{r,r' \in \mathcal{B}_r}([r(\tau) - r(\tau_0)] - [r'(\tau) - r'(\tau_0)]) > 2\epsilon_0$ which means that either $|\tilde{r}(\tau) - \tilde{r}^\star(\tau)| > \epsilon_0$ or $|\tilde{r}'(\tau) - \tilde{r}^\star(\tau)| > \epsilon_0$. Suppose that there are $K$ trajectories which require querying comparison oracle. Suppose that the dataset is composed of

$$\mathcal{D} = \{(\hat{r}_1, \tau_1), \cdots, (\hat{r}_K, \tau_K)\},$$

and $\tilde{r}_1, \ldots, \tilde{r}_K \in \overline{\mathcal{R}}$ are the functions that satisfy $|\tilde{r}_k(\tau_k) - \tilde{r}^\star(\tau_k)| > \epsilon_0$. We now verify that $(\tilde{r}_1, \tau_1, \tilde{r}_2, \tau_2, \cdots, \tilde{r}_K, \tau_K)$ is an eluder sequence (with respect to function class $\overline{\mathcal{R}}$).

The confidence set condition implies

$$\sum_{k<i} (\tilde{r}_i(\tau_k) - \hat{r}_k)^2 \leq \beta.$$

With probability $1 - \delta$, $\forall k \leq K \wedge 2d$, $|\hat{r}_k - \widetilde{r}^{\star}(\tau_k)| \leq \frac{\epsilon_0}{4\sqrt{d}}$ (by Lemma B.5). Then for any $i \leq k$

$$\sum_{k \leq i} (\widetilde{r}_i(\tau_k) - \widetilde{r}^{\star}(\tau_k))^2 \leq \sum_{k \leq i} (\widetilde{r}_i(\tau_k) - \hat{r}_k)^2 + \sum_{k \leq i} (\widetilde{r}_i(\tau_k) - \hat{r}_k + \widetilde{r}_i(\tau_k) - \widetilde{r}^{\star}(\tau_k)) \cdot (\hat{r}_k - \widetilde{r}^{\star}(\tau_k))$$

$$\leq \beta + 2\sqrt{K\beta} \cdot \frac{\epsilon_0}{4\sqrt{d}} + K \left( \frac{\epsilon_0}{4\sqrt{d}} \right)^2 \leq \epsilon_0^2,$$

as long as $K \leq 2d$. In other words, with probability $1 - \delta$, $(\widetilde{r}_1, \tau_1, \widetilde{r}_2, \tau_2, \cdots, \widetilde{r}_{K \wedge 2d}, \tau_{K \wedge 2d})$ is an eluder sequence, which by Definition 2 cannot have length more than $d := \dim_{\mathrm{E}}(\overline{\mathcal{R}}, \epsilon_0)$. It follows that $K \leq \dim_{\mathrm{E}}(\overline{\mathcal{R}}, \epsilon_0)$. $\qquad\square$

**Lemma B.7.** *With probability $1 - \delta$, $r^{\star} \in \mathcal{B}_r$ throughout the execution of Algorithm 1.*

*Proof.* By Lemma B.5 and Lemma B.6, with probability $1 - \delta$, at every step

$$\sum_{(\hat{r}, \tau) \in \mathcal{D}} (r^{\star}(\tau) - r^{\star}(\tau_0) - \hat{r})^2 \leq d \cdot \left( \frac{\epsilon_0}{4\sqrt{d}} \right)^2 \leq \frac{\epsilon_0^2}{4} = \beta.$$

$\qquad\square$

**Lemma B.8.** *With probability $1 - \delta$, for each $\tau$ in Line 3 of Algorithm 1, the returned reward $\hat{r}$ satisfies*

$$|\hat{r} - (r^{\star}(\tau) - r^{\star}(\tau_0))| \leq 2\epsilon_0.$$

*Proof.* We already know that this is true for $\tau$ such that the comparison oracle is queried. However, if it is not queried, then

$$\max_{r, r' \in \mathcal{B}_r} (r(\tau) - r(\tau_0) - (r'(\tau) - r'(\tau_0))) < 2\epsilon_0.$$

Since $r^{\star} \in \mathcal{B}_r$ (by Lemma B.7), this immediately implies $|\hat{r} - (r^{\star}(\tau) - r^{\star}(\tau_0))| \leq 2\epsilon_0$. $\qquad\square$

*Proof of Theorem 4.* Choose $\epsilon_0 := g(\epsilon)/2$, $\beta = \frac{\epsilon_0^2}{4}$ and $m = \Theta(\frac{d_{\overline{\mathcal{R}}} \ln(d_{\overline{\mathcal{R}}}/\delta)}{\epsilon_0^2 \alpha^2})$.

By Lemma B.8 (rescaling $\delta$), with probability $1 - \delta$, the reward returned by the reward interface is $g(\epsilon)$-close to $\widetilde{r}^{\star} := r^{\star} - r^{\star}(\tau_0)$ throughout the execution of the algorithm. By the definition of sample complexity, with probability $1 - \delta$, the policy returned by $\mathscr{A}$ is $\epsilon$-optimal for $\widetilde{r}^{\star}$, which implies that it is also $\epsilon$-optimal for $r^{\star}$. The number of samples (episodes) is bounded by $\mathcal{C}(\epsilon, \delta)$. Finally by Lemma B.6, the number of queries to the comparison oracle is at most

$$\dim_{\mathrm{E}}(\overline{\mathcal{R}}, \epsilon_0) \cdot m \leq \widetilde{\mathcal{O}} \left( \frac{d_{\overline{\mathcal{R}}}^2}{g^2(\epsilon) \alpha^2} \right).$$

$\qquad\square$

## C  P-OMLE: Improved query complexity via white-box modification

While the P2R interface provides a clean and effortless recipe for modifying standard RL algorithms to work with preference feedback, it incurs a large query cost to the comparison oracle, e.g., cubic dependence on $d_{\overline{\mathcal{R}}}$ in Example 3. We believe that this disadvantage is caused by the black-box nature of P2R, and that better query complexities can be achieved by modifying standard RL algorithms in a white-box manner and specialized analysis. In this section, we take OMLE [Liu et al., 2022a] as an example and introduce a standalone adaptation to trajectory preference feedback with improved query complexity. We expect other optimism-based algorithms (e.g., UCBVI-BF and GOLF) can be modified in a similar white-box manner to achieve better query complexity.

The details of the Preference-based OMLE algorithm (P-OMLE) is provided in Algorithm 2. Compared to OMLE with reward feedback (Algorithm 3), the only difference is that now the reward confidence set is computed using preference feedback directly using log-likelihood. Denote by $V_{r,p}^{\pi}$ the expected cumulative reward the leaner will receive if she follows policy $\pi$ in model $(p, r)$. In the $t$-th iteration, the algorithm follows the following steps:

- **Optimistic planning:** Find the policy-model pair $(\pi, r, p)$ that maximizes the value function $V_{r,p}^{\pi}$;

- **Data collection:** Construct an exploration policy set $\Pi_{\exp}(\pi^t)$ [4] and collect trajectories by running all policies in $\Pi_{\exp}(\pi^t)$;

- **Confidence set update:** Update the confidence set using the updated log-likelihood:

$$\mathcal{L}(r, \mathcal{D}_{\mathtt{rwd}}) := \sum_{(\tau, \tau', y) \in \mathcal{D}_{\mathtt{rwd}}} \ln \sigma(r(\tau) - r(\tau'), y), \quad \mathcal{L}(p, \mathcal{D}_{\mathtt{trans}}) := \sum_{(\pi, \tau) \in \mathcal{D}_{\mathtt{trans}}} \ln \mathbb{P}_p^{\pi}(\tau).$$

---

**Algorithm 2** Preference-based OMLE (P-OMLE)

---

1: $\mathcal{B}^1 \leftarrow \mathcal{R} \times \mathcal{P}$
2: execute an arbitrary policy to collect trajectory $\tau^0$
3: **for** $t = 1, \ldots, T$ **do**
4:     compute $(\pi^t, r^t, p^t) = \arg \max_{\pi, (r,p) \in \mathcal{B}^t} [V_{r,p}^{\pi} - r(\tau^0)]$
5:     execute $\pi^t$ to collect a trajectory $\tau$
6:     invoke comparison oracle on $\tau$ and $\tau^0$ to get $y$, add $(\tau, \tau^0, y)$ into $\mathcal{D}_{\mathtt{rwd}}$
7:     **for** each $\pi \in \Pi_{\exp}(\pi^t)$ **do**
8:         execute $\pi$ to collect a trajectory $\tau$, add $(\pi, \tau)$ into $\mathcal{D}_{\mathtt{trans}}$
9:     update

$$\mathcal{B}^{t+1} \leftarrow \big\{ (r, p) \in \mathcal{R} \times \mathcal{P} : \ \mathcal{L}(r, \mathcal{D}_{\mathtt{rwd}}) > \max_{r' \in \mathcal{R}} \mathcal{L}(r', \mathcal{D}_{\mathtt{rwd}}) - \beta_{\mathcal{R}}$$
$$\text{and } \mathcal{L}(p, \mathcal{D}_{\mathtt{trans}}) > \max_{p' \in \mathcal{P}} \mathcal{L}(p', \mathcal{D}_{\mathtt{trans}}) - \beta_{\mathcal{P}} \big\}$$

---

We have the follwoing guarantee for Algorithm 2.

**Theorem C.1.** *Suppose Assumption, 1, 2 and 3 hold. There exists an absolute constant $c_1, c_2 > 0$ such that for any $(T, \delta) \in \mathbb{N} \times (0, 1]$ if we choose $\beta_{\mathcal{R}} = c_1 \ln(|\mathcal{R}|T/\delta)$ and $\beta_{\mathcal{P}} = c_1 \ln(|\mathcal{M}|T/\delta)$ in Algorithm 2, then with probability at least $1 - \delta$, we have that for all $t \in [T]$,*

$$\sum_{t=1}^{T} [V^{\star} - V^{\pi^t}] \leq 2H\xi(d_{\mathcal{P}}, T, c_2\beta_{\mathcal{P}}, |\Pi_{\exp}|) + \mathcal{O}(H\alpha^{-1}\sqrt{d_{\overline{\mathcal{R}}}T\beta_{\mathcal{R}}})$$

*where $d_{\overline{\mathcal{R}}} = \dim_{\mathrm{E}}(\overline{\mathcal{R}}, 1/\sqrt{T})$.*

For problems that satisfy $\xi = \widetilde{\mathcal{O}}(\sqrt{d_{\mathcal{P}}\beta_{\mathcal{P}}|\Pi_{\exp}|T})$, Theorem C.1 implies both the sample complexity and the query complexity for learning an $\epsilon$-optimal policy is upper bounded by:

$$\widetilde{\mathcal{O}}\left( \frac{H^2 d_{\mathcal{P}}|\Pi_{\exp}|^2 \ln |\mathcal{P}|}{\epsilon^2} + \frac{H^2 d_{\overline{\mathcal{R}}} \ln |\mathcal{R}|}{\alpha^2 \epsilon^2} \right).$$

Compared to the result derived through the P2R interface (Example 3), the sample complexity here is basically the same while the query complexity has improved dependence on $d_{\overline{\mathcal{R}}}$ (from cubic to linear). Nonetheless, the query complexity here additionally depends on the complexity of learning the transition model, which is not the case in Example 3.

## C.1 Proofs for P-OMLE

The proof of Theorem C.1 largely follows from that of Theorem D.1. We first introduce several useful notations. Denote by $\mathcal{B}_{\mathcal{M}}^t$, $\mathcal{B}_{\mathcal{P}}^t$ the preference, transition confidence set in the $t$-th iteration, which satisfy $\mathcal{B}^t = \mathcal{B}_{\mathcal{P}}^t \times \mathcal{B}_{\mathcal{M}}^t$. Denote the groundtruth transition and preference by $p^{\star}$ and $M^{\star}$. Denote the trajectories generated when running Algorithm 2 by $\tau^1, \cdots, \tau^T$.

Similar to Lemma D.3, we have that the confidence set satisfies the following properties.

**Lemma C.2.** *There exists absolute constant $c_2$ such that under the same condition as Theorem C.1, we have that with probability at least $1 - \delta$: for all $t \in [T]$*

---

[4]The exploration policy set is problem-dependent and can be simply $\{\pi^t\}$ for many settings.

- $(r^\star, p^\star) \in \mathcal{B}^t$,

- $\sum_{t=1}^{T} \max_{p \in \mathcal{B}_\mathcal{P}^t} d_{\mathrm{TV}}(\mathbb{P}_p^{\pi^t}, \mathbb{P}_{p^\star}^{\pi^t}) \le \xi(d_\mathcal{P}, T, c_2\beta_\mathcal{P}, |\Pi_{\exp}|)$,

- $\sum_{i < t} |\sigma(r^t(\tau^i) - r^t(\tau^0)) - \sigma(r^\star(\tau^i) - r^\star(\tau^0))|^2 \le \mathcal{O}(\beta_\mathcal{R})$.

*Proof.* For the first two statements, see the proof of Theorem 3.2 in Liu et al. [2022a]. For the third bullet point, see Liu et al. [2022a, Proposition B.2], which implies that

$$\mathrm{LHS} \le \mathcal{O}(\beta_\mathcal{R} + \ln(|\mathcal{R}|T/\delta)) = \mathcal{O}(\beta).$$

$\square$

*Proof of Theorem C.1.* By using the first relation in Lemma C.2 and the definition of $(\pi^t, r^t, p^t)$,

$$\sum_{t=1}^{T} [V_{r^\star, p^\star}^\star - V_{r^\star, p^\star}^{\pi^t}]$$

$$= \sum_{t=1}^{T} [V_{r^\star, p^\star}^\star - r^\star(\tau^0) + r^\star(\tau^0) - V_{r^\star, p^\star}^{\pi^t}]$$

$$\le \sum_{t=1}^{T} [V_{r^t, p^t}^{\pi^t} - r^t(\tau^0) + r^\star(\tau^0) - V_{r^\star, p^\star}^{\pi^t}]$$

$$= \sum_{t=1}^{T} [V_{r^t, p^t}^{\pi^t} - V_{r^t, p^\star}^{\pi^t}] + \sum_{t=1}^{T} [V_{r^t, p^\star}^{\pi^t} - r^t(\tau^0) + r^\star(\tau^0) - V_{r^\star, p^\star}^{\pi^t}].$$

We can control the first term by the second inequality in Lemma lem:omle-pref

$$\sum_{t=1}^{T} [V_{r^t, p^t}^{\pi^t} - V_{r^t, p^\star}^{\pi^t}] \le 2 \sum_{t=1}^{T} d_{\mathrm{TV}}(\mathbb{P}_{p^t}^{\pi^t}, \mathbb{P}_{p^\star}^{\pi^t}) \le 2H\xi(d_\mathcal{P}, T, c_2\beta_\mathcal{P}, |\Pi_{\exp}|).$$

For the second term, by Azuma-Hoeffding inequality and by combining the second inequality in Lemma C.2 with the regret guarantee for eluder dimension,

$$\sum_{t=1}^{T} [V_{r^t, p^\star}^{\pi^t} - r^t(\tau^0) + r^\star(\tau^0) - V_{r^\star, p^\star}^{\pi^t}]$$

$$\le \sum_{t=1}^{T} |[(r^t(\tau^t) - r^t(\tau^0)] - [r^\star(\tau^t) - r^\star(\tau^0)]| + \mathcal{O}(H\sqrt{T \ln(1/\delta)})$$

$$\le \mathcal{O}(H\alpha^{-1}\sqrt{d_\mathcal{R} T \beta_\mathcal{R}}).$$

$\square$

# D  OMLE with Perturbed Reward

## D.1  Algorithm details and theoretical guarantees

In this section, we modify the optimistic MLE (OMLE) algorithm [Liu et al., 2022a] to deal with unknown reward functions. The adapted algorithm can then be used with Algorithm 1 as described in Example 3. OMLE is a model-based algorithm that requires a model class $\mathcal{P}$ in addition to the reward class $\mathcal{R}$. On a high level, OMLE maintains a joint confidence set $\mathcal{B}$ in $\mathcal{R} \times \mathcal{P}$. In the $t$-th iteration, the algorithm follows the following steps:

- **Optimistic planning:** Find the policy-model pair $(\pi, r, p)$ that maximizes the value function $V_{r,p}^\pi$ which is the expected cumulative reward the leaner will receive if she follows policy $\pi$ in a model with transition $p$ and reward $r$;

- **Data collection:** Construct an exploration policy set $\Pi_{\exp}(\pi^t)$ [5] and collect trajectories by running all policies in $\Pi_{\exp}(\pi^t)$;
- **Confidence set update:** Update the confidence set using the updated log-likelihood.

The main modification we make is the confidence set of $r$, since the original OMLE algorithm assumes that $r$ is known. The pseudocode of our adapted OMLE algorithm is provided in Algorithm 3.

---

**Algorithm 3** Optimistic MLE with $\epsilon'$-Perturbed Reward Feedback

---

1: $\mathcal{B}^1 \leftarrow \mathcal{R} \times \mathcal{P}$
2: Execute an arbitrary policy to collect trajectory $\tau^0$
3: **for** $t = 1, \ldots, T$ **do**
4:    Compute $(\pi^t, r^t, p^t) = \arg\max_{\pi, (r,p)\in\mathcal{B}^t} V_{r,p}^\pi$
5:    Execute $\pi^t$ to collect a trajectory $\tau$, receive reward $\hat{r}$, add $(\tau, \hat{r})$ into $\mathcal{D}_{\mathtt{rwd}}$
6:    **for** each $\pi \in \Pi_{\exp}(\pi^t)$ **do**
7:       Execute $\pi$ to collect a trajectory $\tau$, add $(\pi, \tau)$ into $\mathcal{D}_{\mathtt{trans}}$
8:    Update
$$\mathcal{B}^{t+1} \leftarrow \Big\{(r, p) \in \mathcal{R} \times \mathcal{P} : \max_{(\tau, \hat{r})\in\mathcal{D}_{\mathtt{rwd}}} |r(\tau) - \hat{r}| \leq \epsilon'$$
$$\text{and } \mathcal{L}(p, \mathcal{D}_{\mathtt{trans}}) > \max_{p'\in\mathcal{P}} \mathcal{L}(p', \mathcal{D}_{\mathtt{trans}}) - \beta_\mathcal{P}\Big\}$$

---

In Line 8, the log-likelihood function is defined as

$$\mathcal{L}(p, \mathcal{D}_{\mathtt{trans}}) := \sum_{(\pi,\tau)\in\mathcal{D}_{\mathtt{trans}}} \ln \mathbb{P}_p^\pi(\tau), \tag{2}$$

where $\mathbb{P}_p^\pi(\tau)$ denotes the probability of observing trajectory $\tau$ in a model with transition function $p$ under policy $\pi$. Other algorithmic parameters $T$ and $\beta_\mathcal{P}$ are chosen in the statement of Theorem D.1.

To obtain sample efficiency guarantees for OMLE, the following assumption is needed.

**Assumption 3** (Generalized eluder-type condition). There exists a real number $d_\mathcal{P} \in \mathbb{R}^+$ and a function $\zeta$ such that: for any $(T, \Delta) \in \mathbb{N} \times \mathbb{R}^+$, transitions $\{p^t\}_{t\in[T]}$ and policies $\{\pi^t\}_{t\in[T]}$, we have

$$\forall t \in [T], \sum_{\tau=1}^{t-1} \sum_{\pi\in\Pi_{\exp}(\pi^\tau)} d_{\mathrm{TV}}^2(\mathbb{P}_{p^t}^\pi, \mathbb{P}_{p^\star}^\pi) \leq \Delta \implies \sum_{t=1}^{T} d_{\mathrm{TV}}(\mathbb{P}_{p^t}^{\pi^t}, \mathbb{P}_{p^\star}^{\pi^t}) \leq \xi(d_\mathcal{P}, T, \Delta, |\Pi_{\exp}|).$$

Here $\mathbb{P}_p^\pi$ is the distribution of trajectories under model $p$ and policy $\pi$, while $|\Pi_{\exp}| := \max_\pi |\Pi_{\exp}(\pi)|$ is the largest possible number of exploration policies.

Assumption 3 shares a similar intuition with the pigeon-hole principle and the elliptical potential lemma, which play important roles in the analysis of tabular MDP and linear bandits respectively. In particular, the $\xi$ function measures the worst-case growth rate of the cumulative error and is the central quantity characterizing the hardness of the problem. Liu et al. [2022a] prove that a wide range of RL problems satisfy Assumption 3 with moderate $d_\mathcal{P}, |\Pi_{\exp}|$ and $\xi = \widetilde{\mathcal{O}}(\sqrt{d_\mathcal{P}\Delta|\Pi_{\exp}|K})$, including tabular MDPs, factored MDPs, observable POMDPs and decodable POMDPs (see Liu et al. [2022a] for more details).

**Theorem D.1.** *Suppose Assumption 1 and 3 hold. There exists absolute constant $c_1, c_2 > 0$ such that for any $(T, \delta) \in \mathbb{N} \times (0, 1]$, if we choose $\beta_\mathcal{P} = c_1 \ln(|\mathcal{P}||\Pi_{\exp}|T/\delta)$ in Algorithm 3, then with probability at least $1 - \delta$, we have that for all $t \in [T]$,*

$$\sum_{t=1}^{T} [V^\star - V^{\pi^t}] \leq 2H\xi(d_\mathcal{P}, T, c_2\beta_\mathcal{P}, |\Pi_{\exp}|)$$

$$+ \mathcal{O}\left(\min_{\omega>0}\left\{T\sqrt{d_\mathcal{R}(\omega)}\epsilon' + \min\{d_\mathcal{R}(\omega), T\}H + T\omega\right\}\right) + \mathcal{O}(H\sqrt{T\ln\delta^{-1}})$$

*where $d_\mathcal{R} = \dim_{\mathrm{E}}(\mathcal{R}, \omega)$.*

---

[5]The exploration policy set is problem-dependent and can be simply $\{\pi^t\}$ for many settings.

For problems that satisfy $\xi = \widetilde{\mathcal{O}}(\sqrt{d_{\mathcal{P}}\beta_{\mathcal{P}}|\Pi_{\exp}|T})$, Theorem D.1 implies a sample complexity of

$$\widetilde{\mathcal{O}}\left(\frac{H^2 d_{\mathcal{P}}|\Pi_{\exp}|^2 \ln|\mathcal{P}|}{\epsilon^2} + \frac{H d_{\mathcal{R}}|\Pi_{\exp}|}{\epsilon}\right).$$

for learning an $\mathcal{O}(\epsilon)$-optimal policy, if we have $\omega = \epsilon$ and $\epsilon' = \epsilon/\sqrt{d_{\mathcal{R}}}$. The sample complexity for specific tractable problems can be found in Appendix D.2.

## D.2  Examples satisfying generalized eluder-type condition

In this section, we present several canonical examples that satisfy the generalized eluder-type condition with $\xi = \widetilde{\mathcal{O}}(\sqrt{d_{\mathcal{P}}\beta_{\mathcal{P}}|\Pi_{\exp}|T})$. More examples can be found in Liu et al. [2022a].

**Example 6** (Finite-precision Factored MDPs). In factored MDPs, each state $s$ consists of $m$ factors denoted by $(s[1], \cdots, s[m]) \in \mathcal{X}^m$. The transition structure is also factored as

$$\mathbb{P}_h(s_{h+1}|s_h, a_h) = \prod_{i=1}^{m} \mathbb{P}^i(s_{h+1}[i]|s_h[\mathrm{pa}_i], a_h),$$

where $\mathrm{pa}_i \subseteq [m]$ is the *parent set* of $i$. The reward function is similarly factored:

$$r_h(s) := \sum_{i=1}^{m} r_h^i(s[i]).$$

Define $B := \sum_{i=1}^{m} |\mathcal{X}|^{\mathrm{pa}_i}$. Factored MDPs satisfy (with $|\Pi_{\exp}| = 1$ and $d_{\mathcal{P}} = m^2|\mathcal{A}|^2 B^2 \mathrm{poly}(H)$)

$$\xi(d_{\mathcal{P}}, T, \Delta, |\Pi_{\exp}|) \leq \sqrt{d_{\mathcal{P}}\Delta T} + AB^2\mathrm{poly}(H).$$

Moreover $\ln|\mathcal{P}| \leq mbAB$, $\ln|\mathcal{R}| \leq mb|\mathcal{X}|$, where $b$ is the number of bits needed to specify each entry of $\mathbb{P}(s_{h+1}||s_h, a_h)$ or $r_h(s)$ [6]. Therefore Theorem D.1 implies a sample complexity of

$$\mathrm{poly}(H) \cdot \widetilde{\mathcal{O}}\left(\frac{bm^3|\mathcal{A}|^3 B^3}{\epsilon^2} + \frac{bm^2|\mathcal{X}|}{\alpha^2\epsilon^2}\right).$$

To proceed, we define partially observable Markov decision process (POMDPs).

**Definition 6.** *In a POMDP, states are hidden from the learner and only observations emitted by states can be observed. Formally, at each step $h \in [H]$, the learner observes $o_h \sim \mathbb{O}_h(\cdot \mid s_h)$ where $s_h$ is the current state and $\mathbb{O}_h(\cdot \mid s_h) \in \Delta_{\mathcal{A}}$ is the observation distribution conditioning on the current state being $s_h$. Then the learner takes action $a_h$ and the environment transitions to $s_{h+1} \sim \mathbb{P}_h(\cdot \mid s_h, a_h)$.*

Liu et al. [2022a] prove that the following subclasses of POMDPs satisfy Assumption 3 with moderate $d_{\mathcal{P}}$ and $|\Pi_{\exp}|$.

**Example 7** ($\alpha$-observable POMDPs). We say a POMDP is $\alpha$-observable if for every $\mu, \mu' \in \Delta_{\mathcal{S}}$,

$$\min_h \|\mathbb{E}_{s\sim\mu}[\mathbb{O}_h(\cdot \mid s)] - \mathbb{E}_{s\sim\mu'}[\mathbb{O}_h(\cdot \mid s)]\|_1 \geq \alpha\|\mu - \mu'\|_1.$$

Intuitively, the above relation implies that different state distributions will induce different observation distributions, and the parameter $\alpha$ measures the amount of information preserved after mapping states to observations. It is proved that $\alpha$-observable POMDPs satisfy Condition 3 with $\Pi_{\exp}(\pi) = \{\pi\}$ and $d_{\mathcal{P}} = \mathrm{poly}(S, A, \alpha^{-1}, H)$ [Liu et al., 2022a].

For simplicity of notations, let $u(h) = \max\{1, h - m + 1\}$.

**Example 8** ($m$-step decodable POMDPs). We say a POMDP is $m$-step decodable if there exists a set of decoder functions $\{\phi_h\}_{h\in[H]}$ such that for every $h \in [H]$, $s_h = \phi_h((o, a)_{u(h):h-1}, o_h)$. In other words, the current state can be uniquely identified from the most recent $m$-step observations and actions. It is proved that $\alpha$-observable POMDPs satisfy Condition 3 with $|\Pi_{\exp}| = A^m$ and $d_{\mathcal{P}} = \mathrm{poly}(L, A^m, H)$ where $L = \max_h \mathrm{rank}(\mathbb{P}_h)$ denotes the rank of the transition matrices $\{\mathbb{P}_h\}_{h\in[H]} \subseteq \mathbb{R}^{SA\times S}$ [Liu et al., 2022a].

---

[6] We can deal with continuous model classes if we use the bracketing number instead of cardinatliy in Theorem D.1

## D.3 Proof of Theorem D.1

We first define some useful notations. Denote by $\mathcal{D}_{\mathrm{rwd}}^t$, $\mathcal{D}_{\mathrm{trans}}^t$ the reward, transition dataset at the end of the $t$-th iteration. We further denote

$$\mathcal{B}_{\mathcal{R}}^t \leftarrow \Big\{ r \in \mathcal{R} : \max_{(\tau, \hat{r}) \in \mathcal{D}_{\mathrm{rwd}}^{t-1}} |r(\tau) - \hat{r}| \leq \epsilon' \Big\},$$

$$\mathcal{B}_{\mathcal{P}}^t \leftarrow \Big\{ p \in \mathcal{P} : \mathcal{L}(p, \mathcal{D}_{\mathrm{trans}}^{t-1}) > \max_{p' \in \mathcal{P}} \mathcal{L}(p', \mathcal{D}_{\mathrm{trans}}^{t-1}) - \beta_{\mathcal{P}} \Big\}.$$

By the definition of $\mathcal{B}^t$, we have that $\mathcal{B}^t = \mathcal{B}_{\mathcal{R}}^t \times \mathcal{B}_{\mathcal{P}}^t$. Denote by $(\tau^t, \hat{r}^t)$ the trajectory-reward pair added into $\mathcal{D}_{\mathrm{rwd}}$ in the $t$-th iteration of Algorithm 3.

By the definition of $\mathcal{B}_{\mathcal{R}}^t$ and the fact that all reward feedback is at most $\epsilon'$-corrupted, we directly have that the confidence set $\mathcal{B}_{\mathcal{R}}^t$ always contain the groundtruth reward function.

**Lemma D.2.** *For all $t \in [T]$, $r^\star \in \mathcal{B}_{\mathcal{R}}^t$.*

Moreover, according to Liu et al. [2022a], the transition confidence set $\mathcal{B}_{\mathcal{P}}^t$ always satisfies the following properties.

**Lemma D.3** (Liu et al. [2022a])**.** *There exists absolute constant $c_2$ such that under Assumption 3 and the same choice of $\beta_{\mathcal{P}}$ as in Theorem D.1, we have that with probability at least $1 - \delta$:*

- $p^\star \in \mathcal{B}_{\mathcal{P}}^t$, *for all $t \in [T]$,*

- $\sum_{t=1}^T \max_{p \in \mathcal{B}_{\mathcal{P}}^t} d_{\mathrm{TV}}(\mathbb{P}_p^{\pi^t}, \mathbb{P}_{p^\star}^{\pi^t}) \leq \xi(d_{\mathcal{P}}, T, c_2 \beta_{\mathcal{P}}, |\Pi_{\exp}|)$.

The first relation states that the transition confidence set contains the groundtruth transition model with high probability. And the second one states that if we use an arbitrary model $p \in \mathcal{B}_{\mathcal{P}}^t$ to predict the transition dynamics under policy $\pi^t$, then the cumulative prediction error over $T$ iterations is upper bounded by function $\xi$.

*Proof of Theorem D.1.* In the following proof, we will assume the two relations in Lemma D.3 hold.

We have that

$$\sum_t \left( V_{r^\star, p^\star}^\star - V_{r^\star, p^\star}^{\pi^t} \right)$$

$$\leq \sum_t \left( V_{r^t, p^t}^{\pi^t} - V_{r^\star, p^\star}^{\pi^t} \right)$$

$$\leq 2H \sum_t d_{\mathrm{TV}}(\mathbb{P}_{p^t}^{\pi^t}, \mathbb{P}_{p^\star}^{\pi^t}) + \sum_t \left( V_{r^t, p^\star}^{\pi^t} - V_{r^\star, p^\star}^{\pi^t} \right)$$

$$\leq 2H \sum_t d_{\mathrm{TV}}(\mathbb{P}_{p^t}^{\pi^t}, \mathbb{P}_{p^\star}^{\pi^t}) + \sum_t \left| r^t(\tau^t) - r^\star(\tau^t) \right| + \mathcal{O}(H\sqrt{T \ln(1/\delta)})$$

$$\leq 2H \xi(d_{\mathcal{P}}, T, c_2 \beta_{\mathcal{P}}, |\Pi_{\exp}|) + \sum_t \left| r^t(\tau^t) - r^\star(\tau^t) \right| + \mathcal{O}(H\sqrt{T \ln(1/\delta)})$$

$$\leq 2H \xi(d_{\mathcal{P}}, T, c_2 \beta_{\mathcal{P}}, |\Pi_{\exp}|) + \mathcal{O}(T\sqrt{d_{\mathcal{R}}} \epsilon' + d_{\mathcal{R}}) + \mathcal{O}(H\sqrt{T \ln(1/\delta)}),$$

where the first inequality uses the definition of $(\pi^t, r^t, p^t)$ and the relation $(r^t, p^t) \in \mathcal{B}^t$, the third one holds with probability at least $1 - \delta$ by Azuma-Hoeffding inequality, the fourth one uses the second relation in Lemma D.3, and the last one invokes the standard regret guarantee for eluder dimension (e.g., Russo and Van Roy [2013]) where $d_{\mathcal{R}} = \dim_{\mathrm{E}}(\mathcal{R}, \epsilon'/2)$. $\qquad\square$

## E Additional details and proofs for Section 4.2

### E.1 Algorithm details

In Algorithm 4, we describe how to learn an approximate von Neumann winner via running two copies of an arbitrary adversarial MDP algorithm. Specifically, we maintain two algorithm instances $\mathscr{A}^{(1)}$ and $\mathscr{A}^{(2)}$. In the $k$-th iteration, we first sample two trajectories $(s_{1:H}^{(1)}, a_{1:H}^{(1)})$ and $(s_{1:H}^{(2)}, a_{1:H}^{(2)})$

without reward by executing $\pi_k^{(1)}$ and $\pi_k^{(2)}$, the two output policies of $\mathscr{A}^{(1)}$ and $\mathscr{A}^{(2)}$, respectively. Then we input $(s_H^{(1)}, s_H^{(2)})$ into the comparison oracle and get a binary feedback $y$. After that, we augment $(s_{1:H-1}^{(1)}, a_{1:H-1}^{(1)})$ with zero reward at the first $H - 2$ steps and reward $y$ at step $H - 1$, which we feed into $\mathscr{A}^{(1)}$. Similarly we create feedback for $\mathscr{A}^{(2)}$ by using $(s_{1:H-1}^{(2)}, a_{1:H-1}^{(2)})$ and reward $1 - y$. The final output policy $\bar{\pi}^{(1)}$ is a uniform mixture of all the policies $\mathscr{A}^{(1)}$ has produced during $K$ iterations.

---

**Algorithm 4** Learning von Neumann Winner via Adversarial MDP Algorithms

---

Initialize two algorithm instances $\mathscr{A}^{(1)}, \mathscr{A}^{(2)}$ for adversarial MDPs with horizon length $H - 1$
**for** $k = 1, \cdots, K$ **do**
    Receive $\pi_k^{(1)}$ from $\mathscr{A}^{(1)}$ and $\pi_k^{(2)}$ from $\mathscr{A}^{(2)}$
    Sample $(s_{1:H}^{(1)}, a_{1:H}^{(1)}) \sim \pi_k^{(1)}$ and $(s_{1:H}^{(2)}, a_{1:H}^{(2)}) \sim \pi_k^{(2)}$
    Query comparison oracle $y \sim \text{Ber}(M(s_H^{(1)}, s_H^{(2)}))$
    Return feedback $(s_1^{(1)}, a_1^{(1)}, 0, \ldots, s_{H-2}^{(1)}, a_{H-2}^{(1)}, 0, s_{H-1}^{(1)}, a_{H-1}^{(1)}, y)$ to $\mathscr{A}_1$
        and $(s_1^{(2)}, a_1^{(2)}, 0, \ldots, s_{H-2}^{(2)}, a_{H-2}^{(2)}, 0, s_{H-1}^{(2)}, a_{H-1}^{(2)}, 1 - y)$ to $\mathscr{A}_2$, respectively
Output average policy mixture $\bar{\pi}^{(1)}$ where $\bar{\pi}^{(1)} := \text{Unif}(\{\pi_k^{(1)}\}_{k \in [K]})$

---

**Converting the output policy to a Markov policy.** Note that the output policy $\bar{\pi}$ is a general non-Markov policy. However, we can convert it to a Markov policy in a sample-efficient manner for tabular MDPs through a simple procedure: execute $\bar{\pi}$ for $N$ episodes, then compute the empirical policy [7]

$$\hat{\pi}_h(a|s) := \frac{J_h(s, a)}{J_h(s)},$$

where $J_h(s, a)$ and $J_h(s)$ denote the visitation counters at state-action pair $(s, a)$ and state $s$ at step $h$, respectively. The following lemma claims that the resulted Markov policy $\hat{\pi}$ is also an approximate restricted Nash equilibrium.

**Lemma E.1.** *If $\bar{\pi}$ is an $\epsilon$-approximate von Neumann winner, then $\hat{\pi}$ is a $2\epsilon$-approximate von Neumann winner with probability at least $1 - \delta$, provided that $N = \widetilde{\Omega}\left(\frac{SA}{\epsilon^2}\right)$.*

### E.2 Proof of Theorem 11

*Proof.* Define $U(\pi, \pi') := \mathbb{E}_{s_H \sim \pi, s'_H \sim \pi'} M[s_H, s'_H]$. The AMDP regret of $\mathscr{A}^{(1)}$ gives

$$\sum_k U\left(\pi_k^{(1)}, \pi_k^{(2)}\right) \geq \max_{\pi: \text{Markov}} \sum_k U\left(\pi, \pi_k^{(2)}\right) - \beta K^{1-c}$$

$$= \max_{\pi: \text{general}} \sum_k U\left(\pi, \pi_k^{(2)}\right) - \beta K^{1-c}$$

where the second inequality uses the fact that there always exists a Markov best response because only the state distribution at the final step matters in the definition of $U$.

Similarly the regret of $\mathscr{A}^{(2)}$ gives

$$\sum_k \left(1 - U\left(\pi_k^{(1)}, \pi_k^{(2)}\right)\right) \geq \max_\pi \sum_k \left[1 - U\left(\pi_k^{(1)}, \pi\right)\right] - \beta K^{1-c}.$$

Summing the two inequalities gives

$$\min_{\pi'} U\left(\bar{\pi}^{(1)}, \pi'\right) \geq \max_\pi U(\pi, \bar{\pi}^{(2)}) - 2\beta K^{1-c}.$$

In other words

$$\text{DGap}(\bar{\pi}^{(1)}, \bar{\pi}^{(2)}) \leq 2\beta / K^c.$$

---

[7]Set $\hat{\pi}_h(\cdot|s) = \text{Unif}(\mathcal{A})$ if $J_h(s) = 0$, *i.e.* if state $s$ is never visited at step $h$.

By the symmetry of preference function, we further have

$$
\begin{aligned}
\mathrm{DGap}(\bar{\pi}^{(1)}, \bar{\pi}^{(1)}) &= \max_{\pi'} U(\pi', \bar{\pi}_1) - \min_{\pi'} U(\bar{\pi}_1, \pi') = 2\max_{\pi'} U(\pi', \bar{\pi}_1) - 1 \\
&= 2\left(\max_{\pi'} U(\pi', \bar{\pi}_1) - U(\bar{\pi}_1, \bar{\pi}_1)\right) \\
&\le 2\mathrm{DGap}(\bar{\pi}^{(1)}, \bar{\pi}^{(2)}) \le 4\beta/K^c.
\end{aligned}
$$

$\square$

### E.3  Details for adversarial linear MDPs.

To apply the regret guarantees from existing works on adversarial linear MDP [e.g., Sherman et al., 2023], we need to show the constructed reward signal in Algorithm 4 is linearly realizable. Since the reward signal is zero for the first $H-2$ steps, we only need to consider step $H-1$. Recall the definition of linear MDPs requires that there exist feature mappings $\phi$ and $\psi$ such that $\mathbb{P}_h(s' \mid s, a) = \langle \phi_h(s, a), \psi(s') \rangle$. By the bilinear structure of transition, the conditional expectation of reward at step $H-1$ in the $k$-th iteration can be written as

$$
\begin{aligned}
\mathbb{E}[y \mid s_{H-1}^{(1)}, a_{H-1}^{(1)}] &= \mathbb{E}[M(s_H^{(1)}, s_H^{(2)}) \mid s_H^{(1)} \sim \mathbb{P}_h(\cdot \mid s_{H-1}^{(1)}, a_{H-1}^{(1)}), \ s_H^{(2)} \sim \pi_k^{(2)}] \\
&= \left\langle \phi_{H-1}(s_{H-1}^{(1)}, a_{H-1}^{(1)}), \sum_{s \in \mathcal{S}} \psi_{H-1}(s)\mathbb{E}[M(s, s_H^{(2)}) \mid s_H^{(2)} \sim \pi_k^{(2)}] \right\rangle.
\end{aligned}
$$

Therefore, the reward function constructed for $\mathscr{A}^{(1)}$ is a linear in the feature mapping $\phi$. Similarly, we can show the reward function constructed for $\mathscr{A}^{(2)}$ is also linear.

### E.4  Proof of Lemma E.1

*Proof.* Let $\iota = c\ln(SAHN/\delta)$ where $c$ is a large absolute constant. By standard concentration, with probability at least $1-\delta$, we have that for all $s$, if $\mathbb{P}^{\bar{\pi}}(s_h = s) \ge \iota/N$, then $J_h(s) \ge N\mathbb{P}^{\bar{\pi}}(s_h = s)/2$. With slight abuse of notation, we define $\bar{\pi}_h(\cdot \mid s_h) = \sum_{a_{1:h-1}, s_{1:h-1}} \bar{\pi}_h(\cdot \mid a_{1:h-1}, s_{1:h})$. Therefore, we have

$$
\begin{aligned}
\sum_s \mathbb{P}^{\bar{\pi}}(s_h = s) \cdot \|\hat{\pi}_h(\cdot \mid s) - \bar{\pi}_h(\cdot \mid s)\|_1 &\le \frac{S\iota}{N} + \sum_s \mathbb{P}^{\bar{\pi}}(s_h = s) \cdot \sqrt{\frac{\iota A}{N\mathbb{P}^{\bar{\pi}}(s_h = s)}} \\
&\le \frac{S\iota}{N} + \sqrt{\frac{\iota SA}{N}} \le \frac{\epsilon}{2H}.
\end{aligned}
$$

Given a Markov policy $\pi_1$ and general policy $\pi_2$, we define

$$
q_h^{\pi_1, \pi_2}(s_h, a_h) := \mathbb{E}_{s_H \sim \pi_1 \mid s_h, a_h, \ s_H' \sim \pi_2}[M(s_H, s_H')].
$$

It follows that for any policy $\pi$,

$$
\begin{aligned}
|U(\hat{\pi}, \pi) - U(\bar{\pi}, \pi)| &= \left| \sum_{h=1}^H \mathbb{E}_{\bar{\pi}}\left[ \langle \hat{\pi}_h(\cdot \mid s_h) - \bar{\pi}_h(\cdot \mid s_h), q_h^{\hat{\pi}, \pi}(s_h, \cdot) \rangle \right] \right| \\
&\le \sum_{h=1}^H \sum_s \mathbb{P}^{\bar{\pi}}(s_h = s) \cdot \|\hat{\pi}_h(\cdot \mid s) - \bar{\pi}_h(\cdot \mid s)\|_1 \le \epsilon/2.
\end{aligned}
$$

Therefore,

$$
\mathrm{DGap}(\hat{\pi}, \hat{\pi}) = 2\max_{\pi'} U(\pi', \hat{\pi}) - 1 \le 2\max_{\pi'} U(\pi', \bar{\pi}) - 1 + \epsilon \le 2\epsilon.
$$

$\square$

# F  Additional details and proofs for Section 4.3

## F.1  Algorithm details

To state the algorithm in a more compact way, we first introduce several notations. We denote the expected winning times of policy $\pi$ against policy $\pi'$ in a RLHF instance with transition $p$ and preference $M$ by

$$V_{p,M}^{\pi,\pi'} := \mathbb{E}\left[M(\tau,\tau') \mid \tau \sim \mathbb{P}_p^\pi,\ \tau' \sim \mathbb{P}_p^{\pi'}\right].$$

Furthermore, we denote the best-response value against policy $\pi$ as

$$V_{p,M}^{\pi,\dagger} = \min_{\pi'} V_{p,M}^{\pi,\pi'},$$

and the minimax value as

$$V_{p,M}^{\star} = \max_{\pi} \min_{\pi'} V_{p,M}^{\pi,\pi'}.$$

---

**Algorithm 5** Learning von Neumann winner via Optimistic MLE

---

1: $\mathcal{B}^1 \leftarrow \mathcal{P} \times \mathcal{M}$
2: execute an arbitrary policy to collect trajectory $\tau^0$
3: **for** $t = 1, \ldots, T$ **do**
4:     compute optimistic von Neumann winner $(\overline{\pi}^t, \overline{M}^t, \overline{p}^t) = \arg\max_{\pi,\ (p,M) \in \mathcal{B}^t} V_{p,M}^{\pi,\dagger}$
5:     compute optimistic best-response $(\underline{\pi}^t, \underline{M}^t, \underline{p}^t) = \arg\min_{\pi',\ (p,M) \in \mathcal{B}^t} V_{p,M}^{\overline{\pi}^t,\pi'}$
6:     sample $\overline{\tau}^t \sim \overline{\pi}^t$ and $\underline{\tau}^t \sim \underline{\pi}^t$
7:     invoke comparison oracle on $(\overline{\tau}^t, \underline{\tau}^t)$ to get $y^t$, add $(\overline{\tau}^t, \underline{\tau}^t, y^t)$ into $\mathcal{D}_{\texttt{pref}}$
8:     **for** each $\pi \in (\Pi_{\exp}(\overline{\pi}^t) \bigcup \Pi_{\exp}(\underline{\pi}^t))$ **do**
9:         execute $\pi$ to collect a trajectory $\tau$, add $(\pi, \tau)$ into $\mathcal{D}_{\texttt{trans}}$
10:    update

$$\mathcal{B}^{t+1} \leftarrow \Big\{(p,M) \in \mathcal{P} \times \mathcal{M} :\ \mathcal{L}(M, \mathcal{D}_{\texttt{pref}}) > \max_{M' \in \mathcal{M}} \mathcal{L}(M', \mathcal{D}_{\texttt{pref}}) - \beta_{\mathcal{M}}$$
$$\text{and } \mathcal{L}(p, \mathcal{D}_{\texttt{trans}}) > \max_{p' \in \mathcal{P}} \mathcal{L}(p', \mathcal{D}_{\texttt{trans}}) - \beta_{\mathcal{P}}\Big\}$$

11: output $\pi^{\text{out}} = \text{Unif}(\{\overline{\pi}^t\}_{t \in [T]})$

---

We provide the pseudocode of learning von Neumann winner via optimistic MLE in Algorithm 5. In each iteration $t \in [T]$, the algorithm performs the following three key steps:

- **Optimistic planning:** Compute the most optimistic von Neumann winner $\overline{\pi}^t$ by picking the most optimistic model-preference candidate $(M, p)$ in the current confidence set $\mathcal{B}^t$. Then compute the most optimistic best-response to $\overline{\pi}^t$, denoted as $\underline{\pi}^t$.

- **Data collection:** Sample two trajectories $\overline{\tau}^t$ and $\underline{\tau}^t$ from $\overline{\pi}^t$ and $\underline{\pi}^t$. Then input them into the comparison oracle to get feedback $y^t$, which is added into the preference dataset $\mathcal{D}_{\texttt{pref}}$. And similar to the standard OMLE, we also execute policies from the exploration policy set that is constructed by using $\overline{\pi}^t$ and $\underline{\pi}^t$, and add the collected data into the transition dataset $\mathcal{D}_{\texttt{trans}}$.

- **Confidence set update:** Update the confidence set using the updated log-likelihood, which is the same as Algorithm 3 except that we replace the utility-basede preference therein by general preference

$$\mathcal{L}(M, \mathcal{D}_{\texttt{pref}}) := \sum_{(\tau,\tau',y) \in \mathcal{D}_{\texttt{pref}}} \ln\left(yM(\tau,\tau') + (1-y)(1 - M(\tau,\tau'))\right).$$

## F.2 Proof of Theorem 12

We first introduce several useful notations. Denote by $\mathcal{B}_\mathcal{M}^t$, $\mathcal{B}_\mathcal{P}^t$ the preference, transition confidence set in the $t$-th iteration, which satisfy $\mathcal{B}^t = \mathcal{B}_\mathcal{P}^t \times \mathcal{B}_\mathcal{M}^t$. Denote the groundtruth transition and preference by $p^\star$ and $M^\star$. To prove Theorem 12, it suffices to bound

$$\sum_t \left( V_{p^\star, M^\star}^\star - V_{p^\star, M^\star}^{\overline{\pi}^t, \dagger} \right).$$

Similar to the proof of Theorem D.1, we first state several key properties of the MLE confidence set $\mathcal{B}^t$, which are trivial extensions of the confidence set properties in Liu et al. [2022a].

**Lemma F.1** (Liu et al. [2022a]). *Under the same condition as Theorem 12, we have that with probability at least $1 - \delta$: for all $t \in [T]$*

- $(p^\star, M^\star) \in \mathcal{B}^t$,

- $\sum_{t=1}^T \max_{p \in \mathcal{B}_\mathcal{P}^t} \left( d_{\mathrm{TV}}(\mathbb{P}_p^{\overline{\pi}^t}, \mathbb{P}_{p^\star}^{\overline{\pi}^t}) + d_{\mathrm{TV}}(\mathbb{P}_p^{\underline{\pi}^t}, \mathbb{P}_{p^\star}^{\underline{\pi}^t}) \right) \leq \xi(d_\mathcal{P}, T, c_2 \beta_\mathcal{P}, |\Pi_{\exp}|)$,

- $\max_{M \in \mathcal{B}_\mathcal{M}^t} \sum_{i < t} |M(\overline{\tau}^i, \underline{\tau}^i) - M^\star(\overline{\tau}^i, \underline{\tau}^i)|^2 \leq \mathcal{O}(\beta_\mathcal{M})$.

The first relation states that confidence set $\mathcal{B}^t$ contains the groundtruth transition-preference model with high probability. The second one resembles the second relation in Lemma D.3. The third one states that any preference model $M$ in confidence set $\mathcal{B}_\mathcal{M}^t$ can well predict the preference over the previously collected trajectory pairs.

By using the first relation in Lemma F.1 and the definition of $\overline{\pi}^t$ and $\underline{\pi}^t$, we have

$$\sum_t \left( V_{p^\star, M^\star}^\star - V_{p^\star, M^\star}^{\overline{\pi}^t, \dagger} \right)$$

$$\leq \sum_t \left( V_{\overline{p}^t, \overline{M}^t}^{\overline{\pi}^t, \dagger} - V_{\underline{p}^t, \underline{M}^t}^{\overline{\pi}^t, \underline{\pi}^t} \right)$$

$$\leq \sum_t \left( V_{\overline{p}^t, \overline{M}^t}^{\overline{\pi}^t, \underline{\pi}^t} - V_{\underline{p}^t, \underline{M}^t}^{\overline{\pi}^t, \underline{\pi}^t} \right)$$

$$\leq 2 \sum_t \left( d_{\mathrm{TV}}(\mathbb{P}_{\overline{p}^t}^{\overline{\pi}^t}, \mathbb{P}_{p^\star}^{\overline{\pi}^t}) + d_{\mathrm{TV}}(\mathbb{P}_{\overline{p}^t}^{\underline{\pi}^t}, \mathbb{P}_{p^\star}^{\underline{\pi}^t}) + d_{\mathrm{TV}}(\mathbb{P}_{\underline{p}^t}^{\overline{\pi}^t}, \mathbb{P}_{p^\star}^{\overline{\pi}^t}) + d_{\mathrm{TV}}(\mathbb{P}_{\underline{p}^t}^{\underline{\pi}^t}, \mathbb{P}_{p^\star}^{\underline{\pi}^t}) \right)$$

$$+ \sum_t \left( V_{p^\star, \overline{M}^t}^{\overline{\pi}^t, \underline{\pi}^t} - V_{p^\star, \underline{M}^t}^{\overline{\pi}^t, \underline{\pi}^t} \right)$$

$$\leq 2 \sum_t \left( d_{\mathrm{TV}}(\mathbb{P}_{\overline{p}^t}^{\overline{\pi}^t}, \mathbb{P}_{p^\star}^{\overline{\pi}^t}) + d_{\mathrm{TV}}(\mathbb{P}_{\overline{p}^t}^{\underline{\pi}^t}, \mathbb{P}_{p^\star}^{\underline{\pi}^t}) + d_{\mathrm{TV}}(\mathbb{P}_{\underline{p}^t}^{\overline{\pi}^t}, \mathbb{P}_{p^\star}^{\overline{\pi}^t}) + d_{\mathrm{TV}}(\mathbb{P}_{\underline{p}^t}^{\underline{\pi}^t}, \mathbb{P}_{p^\star}^{\underline{\pi}^t}) \right)$$

$$+ \sum_t \left( \overline{M}^t(\overline{\tau}^t, \underline{\tau}^t) - \underline{M}^t(\overline{\tau}^t, \underline{\tau}^t) \right) + \mathcal{O}\left( \sqrt{T \ln(1/\delta)} \right),$$

By the second relation in Lemma F.1 and Definition 3, we have

$$\sum_t \left( d_{\mathrm{TV}}(\mathbb{P}_{\overline{p}^t}^{\overline{\pi}^t}, \mathbb{P}_{p^\star}^{\overline{\pi}^t}) + d_{\mathrm{TV}}(\mathbb{P}_{\overline{p}^t}^{\underline{\pi}^t}, \mathbb{P}_{p^\star}^{\underline{\pi}^t}) + d_{\mathrm{TV}}(\mathbb{P}_{\underline{p}^t}^{\overline{\pi}^t}, \mathbb{P}_{p^\star}^{\overline{\pi}^t}) + d_{\mathrm{TV}}(\mathbb{P}_{\underline{p}^t}^{\underline{\pi}^t}, \mathbb{P}_{p^\star}^{\underline{\pi}^t}) \right) \leq 4\xi(d_\mathcal{P}, T, c\beta_\mathcal{P}, |\Pi_{\exp}|),$$

where $c$ is an absolute constant.

Combining the third relation in Lemma F.1 with the regret bound of eluder dimension (e.g., Lemma 2 in Russo and Van Roy [2013]), we have

$$\sum_t \left( \overline{M}^t(\overline{\tau}^t, \underline{\tau}^t) - \underline{M}^t(\overline{\tau}^t, \underline{\tau}^t) \right) \leq \mathcal{O}(\sqrt{d_\mathcal{M} \beta_\mathcal{M} T}).$$

Putting all pieces together, we have

$$\sum_t \left( V_{p^\star, M^\star}^\star - V_{p^\star, M^\star}^{\overline{\pi}^t, \dagger} \right) \leq 4\xi(d_\mathcal{P}, T, c\beta_\mathcal{P}, |\Pi_{\exp}|) + \mathcal{O}(\sqrt{d_\mathcal{M} \beta_\mathcal{M} T}).$$

