# OpenReview forum: "Is RLHF More Difficult than Standard RL? A Theoretical Perspective"
_NeurIPS.cc/2023/Conference — NeurIPS 2023 poster_

### Official Review · Reviewer_XC4t · 2023-07-07

**Soundness:** 4 excellent
**Presentation:** 4 excellent
**Contribution:** 3 good
**Rating:** 6
**Confidence:** 4

**Summary:**

The authors consider the RLHF setting. First, in the case where there (a) exists a ground-truth utility function and (b) feedback is positive with higher probability when there is a larger difference in rewards, they derive an algorithm to iteratively winnow down a ball of reward functions. When such structure cannot be assumed, they derive a reduction to adversarial MDPS for solving for the von Neumann winner.

**Strengths:**

(+) The paper is clearly written and the algorithms are intuitive and reasonable.

(+) I haven't seen someone consider the von Neumann winner before in the RLHF space.

**Weaknesses:**

(-) I think there's some weird formatting on the top of pg. 8 as the line numbers stop. Could you fix this? Also, I think you're missing a superscript $(i)$ in the definition of a partial trajectory. Also, please add some space between line 343 and the equation below.

(-) There are no experiments whatsoever.

**Questions:**

1) Could you add in https://arxiv.org/pdf/2305.18505.pdf and https://arxiv.org/pdf/2305.14816.pdf to your related work section and discuss how your approach differs from theirs?

2) Do you get any sort of agnostic guarantee for Alg. 1 in the non-realizable case?

3) Are "oracle complexity" (line 169) and "query complexity" (footnote 2) the same thing? If so, could you pick a term and stick to it?

4) Imagine if instead of receiving stochastic feedback, you received a deterministic (+/-) label from the oracle. While this is a simpler setting in some sense, I'm not sure if you'd still be able to compare things to a fixed initial trajectory. Do you have any thoughts on how you could modify Alg. 1 to handle this setting?

5) Could you clarify how you could run Alg. 1 with k-wise comparison feedback? I found section 3.3 of the paper unfortunately vague.

6) Footnote 3's reasoning is a bit artificial. Could you instead motivate the setting you consider by noting that for empirical RLHF, people usually treat it as a "bandit" problem and only give feedback at the ending?

7) Could you run something like NPG instead as a no-regret algorithm for the adversarial MDPs your reduction requires solving? If so, that might be an illustrative point to add in given it's a bit closer to the PPO-style algorithms people use in practical RLHF. Also, given everything is fully observed, is there an algorithm that gets a tighter rate than OMLE? It sort of feels like you're using a heavier hammer than necessary.

---

> ### Author Rebuttal · Authors · 2023-08-08
>
>
> We thank the reviewer for their positive evaluation and detailed feedback. We would address the reviewer’s concerns as follows.
>
> **Q1. Formatting issues.**
>
> **A1.** Thank you for pointing out these issues. We will correct them in the final version since we cannot update the paper during the rebuttal process.
>
> **Q2. Additional related work.**
>
> **A2.** Thank you for bringing these two papers to our attention. We will revise the related work section accordingly in the final version.
>
> **Q3. “Do you get any sort of agnostic guarantee for Alg. 1 in the non-realizable case?”**
>
> **A3.** If the ground-truth reward function is not realizable but known to have a small approximation error, i.e. $\Vert r^\star - r\Vert_\infty<\Delta$ for some $r\in \mathcal{R}$, we believe it is possible to obtain results on learning the optimal policy up to error $O(\sqrt{d_r} \Delta)$, where the approximation error is amplified by the square root of the eluder dimension similar to the case in bandits with misspecification (Lattimore et al., 2020).
>
> **Q4. On “oracle complexity” and “query complexity”.**
>
> **A4.** Yes, they are referring to the same concept. We will modify Line 169 to “query complexity”.
>
> **Q5. On deterministic labels.**
>
> **A5.** If labels are deterministic in the utility-based setting, we have established an impossibility result (Lemma 3) where it is impossible to identify the optimal policy. Intuitively, this is because when labels are deterministic, only ordinal information about the reward function is kept and the rest is lost. For instance, one can only learn that $\tau_1$ is preferred over $\tau_2$ and $\tau_2$ over $\tau_3$. However, without information on the actual reward differences, one cannot compare a policy that generates $\tau_2$, and one that generates $\tau_1$ with probability 0.5 and $\tau_3$ with probability 0.5.
>
> On the other hand, in the non utility-based setting, our results can handle deterministic or arbitrary link functions. However, the solution concept would be different from that of the utility-based setting.
>
> **Q6. Clarification on k-wise comparison.**
>
> **A6.** To adapt Algorithm 1 to k-wise comparison, we simply need to change Line 7 from
>
> > Query comparison oracle $m$ times on $\tau$ and $\tau_0$; compute average comparison result $\bar{o}$
>
> to
>
> > For $i$ from 1 to $\lceil 2m/k\rceil$:
> > - Query comparison oracle on $(\tau,\tau_0,\cdots,\tau,\tau_0)$, receive output $\phi$
> > - Compute $o_i\gets \frac{2}{k}\sum_{j=1}^{k/2} I[\phi(2j-1)>\phi(2j)]$
> >
> > Compute the average $\bar{o}\gets \frac{1}{\lceil 2m/k\rceil} \sum_{i=1}^{\lceil 2m/k\rceil} o_i$
>
> Here, instead of querying a two-way comparison oracle for $m$ times, we alternatively query a $k$-wise comparison oracle for $\lceil 2m/k\rceil$ times. By Proposition 5, the computed $\bar{o}$ has the same statistical property as if using $k/2 \cdot \lceil 2m/k\rceil$ independent samples from a two-way comparison oracle.
>
>
> **Q7. On NPG as an adversarial MDP solver.**
>
> **A7.** There are policy optimization style algorithms for adversarial MDPs in the tabular setting (see e.g. Algorithm 3 in Efroni et al. 2020), which could indeed be used as the solver required by the reduction in Section 4.2.
>
> **Q8. "Also, given everything is fully observed, is there an algorithm that gets a tighter rate than OMLE? It sort of feels like you're using a heavier hammer than necessary."**
>
> **A8.** In certain cases like tabular MDPs and kernel linear MDPs, we believe it is possible to obtain sharper rates by replacing the MLE-based confidence sets with ones that are specially designed to exploit the problem structures. For example, we can use Bernstein inequality to construct tighter confidence intervals for each entry $P(s’ \mid s,a)$ of the transition matrix in the tabular setting. Nonetheless, the current version uses OMLE-style algorithms because of its algorithmic simplicity and its generality. Specifically, we can use one single algorithm (OMLE) to provide polynomial sample-efficiency guarantees for many distinctive RLHF problems, e.g., tabular MDPs, factored MDPs, linear kernel MDPs etc. Moreover, it also directly generalizes to RLHF under partial observability, e.g., observable POMDPs and decodable POMDPs.
>
> ---
>
> Lattimore et al. Learning with Good Feature Representations in Bandits and in RL with a Generative Model. 2020.
>
> Efroni et al. Optimistic Policy Optimization with Bandit Feedback. 2020.
>
> Lin Yang, Mengdi Wang. Reinforcement Learning in Feature Space: Matrix Bandit, Kernels, and Regret Bound. 2020.

---

> > ### Comment · Reviewer_XC4t · 2023-08-11
> > **Re:**
> >
> > I thank the authors for their rebuttal. After reading it and the comments of the other reviewers, I would be most comfortable with keeping my score where it is.
> >
> > Please remember to make the changes you promised to above! I would also add in some of the text from A6/A7 to the main paper, as well as updating Footnote 3, when you have the chance.

---

### Official Review · Reviewer_ZkeP · 2023-07-07

**Soundness:** 2 fair
**Presentation:** 1 poor
**Contribution:** 3 good
**Rating:** 4
**Confidence:** 3

**Summary:**

Refined writing: The objective of this paper is to establish a theoretical foundation for reinforcement learning based on human feedback preferences. The authors conduct an analysis on two aspects: (1) utility-based preferences in tabular MDPs, linear MDPs, and MDPs with low Bellman-Ruler dimension, and (2) general preferences considering the assumption of the calculation of von Neumann winners can be reduced to finding NEs of games with independent transition dynamics.

**Strengths:**

Refined writing: The current limitations in the theoretical understanding of reinforcement learning from human feedback have prompted the need for addressing crucial questions, particularly in light of the significant success achieved by RLHF in applications such as ChatGPT. This paper effectively tackles these important questions, contributing to a deeper understanding of the subject matter.

**Weaknesses:**

The research topic of this paper is indeed interesting, and the reviewer initially had high expectations. However, there is a significant discrepancy between the paper's title and the actual content presented in the theoretical analyses.

The first part of analyses primarily focus on the utility-based preferences, which is applicable to simple linear MDPs or MDPs with lower Bellman-Ruler dimensions. It is challenging to extend the conclusion that "human query feedback does not scale with the sample complexity of the reinforcement learning algorithm" to more powerful function approximators commonly used in complex learning settings. Additionally, it raises the question of whether human feedback is even necessary for learning in these simple MDPs. The reviewer thinks that high-cost factors like human feedback are typically introduced in sophisticated tasks such as large language models and autonomous driving.

Certain sections of the paper, particularly the discussion on general preferences, are challenging to follow. There is a lack of exploration regarding how the theoretical analyses and computation of the von Neumann winner can be translated into an algorithm for relating preferences to rewards. This omission makes it difficult to assess the practicality and applicability of the proposed method, as it is based on the assumption of factorizable underlying Markov games. In realistic scenarios, independent transition dynamics are not commonly observed. This raises concerns about the completeness of this section in the paper.

If the results of the paper are limited to simple MDPs, the authors may consider changing the title to reflect this specificity.

Based on the reviewer's assessment, the current state of the paper does not warrant immediate publication. The authors are encouraged to reorganize their work and submit a more comprehensive version to top machine learning venues in the future.

**Questions:**

Please find the questions in the weakness section.

**Limitations:**

This paper didn't discuss limitations of their analyses in a separate section, but it clearly states their assumptions when introducing the theoretical findings.

---

> ### Author Rebuttal · Authors · 2023-08-08
>
> We thank the reviewer for their comments, and we will address them below.
>
> **Q1.** The first part of analyses primarily focus on the utility-based preferences, which is applicable to simple linear MDPs or MDPs with lower Bellman-Ruler dimensions. It is challenging to extend the conclusion that "human query feedback does not scale with the sample complexity of the reinforcement learning algorithm" to more powerful function approximators commonly used in complex learning settings.
>
> **A1.** Theorem 4 shows that the query complexity of P2R Interface only scales with the complexity of learning the reward function and is independent of the complexity of the RL algorithm, regardless of the RL tasks and algorithms we are considering. As a result, even in a complex learning setting with powerful function approximators, the query complexity of P2R Interface is still independent of the RL algorithms. Nonetheless, P2R Interface may have larger query complexity in more complex tasks because the reward function could be harder to learn.
>
> **Q2.**  Additionally, it raises the question of whether human feedback is even necessary for learning in these simple MDPs.
>
> **A2.** This paper studies the setting of RLHF where human feedback is the only information available and rewards either are not observable (utility-based setting) or may not exist (non utility-based setting). Therefore, learning is impossible without using human feedback in the setting of RLHF.
>
> **Q3.**  There is a lack of exploration regarding how the theoretical analyses and computation of the von Neumann winner can be translated into an algorithm for relating preferences to rewards.
>
> **A3.**  Experiments have shown that human preferences can be intransitive [Tversky, 1969], which implies there is NO reward function that can represent human preference in certain tasks. As a result, we adopt a new solution concept—von Neumann winner, and provide two schemes to learn it: (1) reduce learning von Neumann winner to learning factorizable Markov games and then apply adversarial MDP algorithms; (2) a preference-based version of the OMLE algorithm. Finally, we remark that it is impossible to relate preferences to rewards because no reward function can represent intransitive preference.
>
> **Q4.**  This omission makes it difficult to assess the practicality and applicability of the proposed method, as it is based on the assumption of factorizable underlying Markov games. In realistic scenarios, independent transition dynamics are not commonly observed.
>
> **A4.** We neither assume factorizable Markov games nor assume independent transition dynamics. Instead, we reduce the problem of finding von Neumann winners to solving factorizable Markov games (Proposition 9). Moreover, the reduction holds for any RLHF problems without any assumption.

---

> > ### Comment · Reviewer_ZkeP · 2023-08-18
> > **Thanks for your response**
> >
> > The authors' response has effectively addressed many concerns raised by the reviewer. Nevertheless, the reviewer still thinks there is a disparity between the paper's title and the theorems presented within.  While it is true that the query complexity of the proposed Interface remains independent of the choice of RL algorithms, it is intrinsically linked to the intricacy of the reward function. In cases beyond linear MDPs or MDPs with lower Bellman-Ruler dimensions, the reward function is expected to exhibit complexity, potentially resulting in poor scalability of the presented conclusions. In contrast, the title imparts the impression that the authors have resolved this pivotal question. The motivating examples in the paper is on recommendation systems, image generation, robotics, and large language models. It is unlikely that MDPs are linear here.
> >
> > Moreover, in the simple MDP cases, the authors put that "This paper studies the setting of RLHF where human feedback is the only information available and rewards either are not observable (utility-based setting) or may not exist (non utility-based setting)". Is such a setting common in real-world applications? Why do we even care about this setting? Why not directly design a reward function for these MDPs, which is not impossible given that the underlying MDP is not complex.
> >
> > In light of these considerations, the reviewer's rating has been adjusted to "4". The reviewer maintains the viewpoint that this paper poses an intriguing question, acknowledges the inherent challenges in conducting theoretical analyses, and commends the efforts invested in addressing this question. However, as it stands, the paper appears to be unprepared for publication. The authors could also enhance the writing, potentially by focusing solely on presenting pivotal and insightful findings in the main paper.

---

> > > ### Author Response · Authors · 2023-08-18
> > > **Further clarification**
> > >
> > > We thank the reviewer for their time and effort, and we are glad that our previous rebuttal has addressed many of their concerns. We would like to make two further clarifications here.
> > >
> > > **1. Complexity of the Underlying MDP:**
> > >
> > > > The motivating examples in the paper are on recommendation systems, image generation, robotics, and large language models. It is unlikely that MDPs are linear here.
> > >
> > > We wish to emphasize that our main theoretical result does not in any way assume linearity of the MDP. Instead, the guarantees hold as long as there exist efficient reward-based RL algorithms. Low Bellman-Eluder dimension MDPs are simply an example where provably efficient RL algorithms are known. Therefore, the complexity of the MDPs for recommendation systems, image generation, robotics, and large language models does not detract from the general applicability of our main finding, which is **RLHF is not statistically harder than standard RL**.
> > >
> > > **2. Relevance of the RLHF Setting:**
> > >
> > > > The authors put that "This paper studies the setting of RLHF where human feedback is the only information available and rewards either are not observable (utility-based setting) or may not exist (non utility-based setting)". Is such a setting common in real-world applications? Why do we even care about this setting?
> > >
> > > Indeed, the setting we explore is prevalent in a myriad of real-world applications. For instance, in scenarios like tuning language models or improving human-robot interactions, manually designing a reward function a priori is difficult if not impossible, which makes *learning* a reward function from human feedback extremely attractive. Exploiting human feedback sidesteps the challenges of manually specifying a reward function, which can be error-prone and might not fully capture nuanced human preferences. This is precisely the key motivation of RLHF. The emphasis on RLHF is driven by its widespread utility in these and other applications, making our study both relevant and timely.
> > >
> > >
> > > We appreciate the reviewer's insightful feedback and hope these clarifications further elucidate the novelty and applicability of our work. We remain committed to refining our manuscript to better convey our findings and address any ambiguities.

---

> > > > ### Comment · Reviewer_ZkeP · 2023-08-18
> > > > **Thanks**
> > > >
> > > > I feel most at ease maintaining my current score. It's possible that the authors might consider improving the writing of their paper to enhance its readability, rather than allowing it to appear hastily completed.

---

### Official Review · Reviewer_nikD · 2023-07-09

**Soundness:** 4 excellent
**Presentation:** 3 good
**Contribution:** 4 excellent
**Rating:** 7
**Confidence:** 3

**Summary:**

The authors study the problem of learning in preference-based RL and investigate the question of whether preference based RL is any harder that reward based RL. They show that for preferences that are based on an underlying reward function, preference based RL is no harder than reward based RL for most of the theoretical RL settings including tabular MDPs, linear MDPs, MDPs with finite eluder dimension. They propose an efficient algorithm that can solve this problem by reducing it to the reward based RL and solving the reward version instead. Further, they demonstrate that this reduction does not incur any additional sample complexity, requiring humans to provide preference on only a small portion of trajectories collected by the algorithm.

For general preference function, they propose a framework of factored Multi-Agent MDP to find the “von Neumann winner” solution concept. They further provide algorithms based on Adversarial MDP and Optimistic MLE to solve the problem in the setting where preference are entirely based on last state and on complete trajectory respectively.

**Strengths:**

- The paper is well written and except minor notation issues it is a good read.

- The authors provide algorithms to solve the preference based RL problem with a theoretical guarantee on sample and computation complexity in different RL settings including finite state, linear and bounded eluder dimension settings.

- The authors also study this problem in general preference feedback setting and propose the solution concept of “von Neumann winner”. They further propose algorithms when preference depend  on the final state or entirety of the two compared trajectories.

**Weaknesses:**

- The paper assumes that humans can provide preferential feedback at trajectory level which is not always easy.
- The concept of Eluder dimension and following results are not very intuitive.
- The paper lacks any experimental results but the theoretical contribution is strong enough.

**Questions:**

1. In line 116-117 : Why is underlying reward function not state-action Markovian? The corresponding value notation is not correct.
2. Can you compare two states as in line 139 or is it two (state,action) pair?
3. Why the input trajectories needs to be feasible with respect to underlying environment? Where is this assumption used?

**Limitations:**

- The paper does not address the setting when users can only provide preference at state and not trajectory level.

---

> ### Author Rebuttal · Authors · 2023-08-08
>
>
> We thank the reviewer for their positive evaluation and detailed feedback. We would address the reviewer’s questions as follows.
>
> **Q1.** “The paper assumes that humans can provide preferential feedback at trajectory level”
>
> **A1.** We would like to clarify that in the utility-based setting, our results could apply to the setting where human labellers provide feedback on a state-action pair, where the preference is assumed to be based on the immediate reward (see Line 133-136 and Remark 1). We present primarily the results for trajectory-based feedback setting since this is the setting considered in most prior work (see e.g. Pacchiano et al. 2021, Novoseller et al. 2020).
>
> For the non-utility-based setting, we do have results (Section 4.2) on the case where the preference is based on the final-state of the trajectory. Without this assumption, it is highly unclear how one can reason about preferences of trajectory based on human feedback at a state-action pair level.
>
> **Q2.** “In line 116-117 : Why is the underlying reward function not state-action Markovian?”
>
> **A2.** Thank you for pointing out the notational inconsistency. Indeed, in Line 116-117, the reward function $r^\star$ should be state-action based (that is $r^\star = [H]\times \mathcal{S}\times \mathcal{A} \to [0,1]$).
>
> **Q3.** “Can you compare two states as in line 139 or is it two (state,action) pair?”
>
> **A3.** Thank you for pointing out this typo. Line 139 should be modified as “... evaluator prefers $\tau$ over $\tau’$ is”.
>
> **Q4.** “Why does the input trajectories need to be feasible with respect to the underlying environment?”
>
> **A4.** The feasibility requirement is more of a restriction on algorithms (that our algorithms meet) than an assumption. The motivation of this condition is to rule out trivial but impractical strategies. For instance, one can learn the full reward function using *no* samples from the MDP by iteratively querying the trajectory with the highest uncertainty in the whole trajectory space. However, such synthesized trajectories could be random sequences of pixels or streams of incoherent speech, which human labelers cannot reliably evaluate. In other words, the feasibility requirement can be thought of as a weakening of Definition 1: instead of assuming the comparison oracle to be valid on the whole trajectory space, we can alternatively assume that it is only valid on trajectories that are feasible.
>
>
> ---
>
> Aldo Pacchiano, Aadirupa Saha, and Jonathan Lee. Dueling rl: reinforcement learning with trajectory preferences. 2021.
>
> Ellen Novoseller, Yibing Wei, Yanan Sui, Yisong Yue, and Joel Burdick. Dueling Posterior Sampling for Preference-Based Reinforcement Learning. 2020.

---

### Official Review · Reviewer_Sju6 · 2023-07-24

**Soundness:** 2 fair
**Presentation:** 1 poor
**Contribution:** 1 poor
**Rating:** 5
**Confidence:** 1

**Summary:**

The authors attempt to show the conditions under which RLHF is theoretically identical to standard RL where a reward function is specified as a part of the environment. The algorithm P2R Interface is given as way to learn from preference feedback such that all requirements are met for RLHF to be identical to standard RL. The reduction of RLHF to standard RL is discussed in the context of utility-baed preferences and general preferences that do not meet the requirements of a utility function.

**Strengths:**

- The paper includes an extensive related works section.

**Weaknesses:**

- The paper is difficult for me to follow and understand. It is very full of jargon for which no explanation is provided.
- It is not well motivated why the "reduction" the authors propose to make is necessary and the benefits it carries.
- A discussion section to wrap up what has been shown in the paper would be helpful.
- Figure 1 is missing.

**Questions:**

- In the utility-based preferences scenario the authors state that the trajectories given as input to the comparison oracle must generated by the policy. How does this work in the case of Ouyang et al. [2022]'s approach where the reward model is learned in advance and therefore are not generated by the policy?

**Limitations:**

- The paper is non-trivial for someone not an expert on the specific topic of the paper to follow. It would be great for the paper to be more easily understandable by those with a background in preference-based RL, so that they may incorporate the learnings into their work.

---

> ### Author Rebuttal · Authors · 2023-08-08
>
> **Q1.** The paper is difficult for me to follow and understand. It is very full of jargon for which no explanation is provided.
>
> **A1.** We kindly ask the reviewer to specify the sections of the paper they found ambiguous. We are eager to provide clarifications where needed.
>
> **Q2.**  It is not well motivated why the "reduction" the authors propose to make is necessary and the benefits it carries.
>
> **A2.** The term “reduction’’ in computer science refers to a scheme of transforming one problem into another problem. In our paper, the reduction is used to convert the RLHF problem to a standard reward-aware RL problem. The benefits of designing such a reduction (as opposed to a single RLHF algorithm) is that any standard RL algorithm can be combined with the reduction to derive a RLHF algorithm.
>
> **Q3.**  A discussion section to wrap up what has been shown in the paper would be helpful.
>
> **A3.** Thank you for the suggestion. We will add the following conclusion section in the revision:
> This paper studies RLHF via efficient reductions. For utility based preferences, we introduce a Preference-to-Reward Interface which reduces preference based RL to standard reward-based RL. Our results are amenable to function approximation and incur no additional sample complexity. For general preferences without underlying rewards, we reduce finding the von Neumann winner to finding restricted Nash equilibrium in a class of Markov games. This can be more concretely solved by adversarial MDP algorithms if the preference depends solely on the final state, and by optimistic MLE for preferences that depend on the whole trajectory. Our results demonstrate that RLHF, from both utility-based and general preferences, can be readily solved under standard assumptions and by existing algorithmic techniques in RL theory literature. This suggests that RLHF is not much harder than standard RL in the complexity sense, and needs not to be more complicated in the algorithmic sense. Consequently, our findings partially answer our main query: RLHF may not be more difficult than standard RL.
>
> **Q4.**  Figure 1 is missing.
>
> **A4.** Please refer to Appendix A in the supplementary materials for Figure 1.
>
> **Q5.**  In the utility-based preferences scenario the authors state that the trajectories given as input to the comparison oracle must be generated by the policy. How does this work in the case of Ouyang et al. [2022]'s approach where the reward model is learned in advance and therefore are not generated by the policy?
>
> **A5.** Our algorithms (e.g., P2R Interface) only need to query the comparison oracle with trajectories generated by the policies. This means that all results in this paper still hold without any change even if we allow the comparison oracle to compare arbitrary trajectories, as is the case in Ouyang et al. [2022].

---

> > ### Comment · Reviewer_Sju6 · 2023-08-14
> >
> > A1. We kindly ask the reviewer to specify the sections of the paper they found ambiguous. We are eager to provide clarifications where needed.
> > - It would be great to have an example or two comparing the two approaches here with some citations: "These works typically develop specialized algorithms and analysis in a white-box fashion, instead of building on existing techniques in standard RL." What are examples of how they specialized and what examples of the standard RL techniques that should be build upon?
> > - It would be helpful to have more explanation of what is meant by "general (arbitrary) preferences" earlier in the paper.
> > - What is a "confidence set of rewards Br"? Is this an ensemble of reward models and you are checking for ensemble agreement? Does this mean there will likely be more queries to the oracle at early stages of policy/reward training?
> > - Why compare against "a fixed trajectory τ0"? How is the fixed trajectory selected? Does this fixed trajectory need to have some quality guarantees?
> >
> > A4. Please refer to Appendix A in the supplementary materials for Figure 1.
> > - If a main figure for the paper is in the Appendix then that should be called out when the figure is referenced.

---

> > > ### Author Response · Authors · 2023-08-16
> > > **Response to further questions**
> > >
> > > We thank the reviewer for the follow-up response. The questions will be addressed below.
> > >
> > > > It would be great to have an example or two comparing the two approaches here with some citations.
> > >
> > > Examples comparing our approach and that of existing studies can be found in Section 1.1, L85-99. The main difference between our approach and previous works mentioned in L85-99 is that our work is reduction-based and can directly work with existing reward-based RL algorithms.
> > >
> > > > It would be helpful to have more explanation of what is meant by "general (arbitrary) preferences" earlier in the paper.
> > >
> > > The term “general preferences” is used in contrast to “utility-based preferences”, which are briefly explained in Line 45. We will include an additional note in Line 55 in the revision for better clarity.
> > >
> > > > What is a "confidence set of rewards Br"?
> > >
> > > The confidence set of rewards $B_r$ is a set of reward functions that is maintained by the algorithm (see Line 10 of Algorithm 1). For instance, if the reward function class is linear, the confidence set $B_r$ would correspond to an ellipsoid in the parameter space.
> > >
> > > > Does this mean there will likely be more queries to the oracle at early stages of policy/reward training?
> > >
> > > It would be hard to give a definite answer for MDPs in general. The oracle would be queried when the online RL algorithm generates trajectories (or state-action pairs) that are “novel” for the P2R interface. When such trajectories are visited would depend heavily on the exploration strategy used by the online RL algorithm.
> > >
> > > > Why compare against "a fixed trajectory τ0"?
> > >
> > > The fixed trajectory $\tau_0$ used in Algorithm 1 is collected in Line 2 at the start of the algorithm by executing a uniformly random policy. By Definition 1, the comparison oracle returns a result based on the *reward differences” of two trajectories. Therefore by comparing against a fixed trajectory $\tau_0$, we could learn $r(\cdot)-r(\tau_0)$ — the groundtruth reward function up to a fixed offset. The trajectory $\tau_0$ can be an arbitrary trajectory in principle due to Assumption 2. The reason that we used a trajectory generated by a random policy is to meet the feasibility criterion discussed in Line 130.
> > >
> > > > If a main figure for the paper is in the Appendix then that should be called out when the figure is referenced.
> > >
> > > We will clarify that Figure 1 is in Appendix A in references to it.

---

> > > > ### Comment · Reviewer_Sju6 · 2023-08-18
> > > >
> > > > Thank you for your responses.
> > > >
> > > > Based on your feedback and the other reviews, I have changed by score to 5. I do agree with Reviewer ZkeP on, "The authors could also enhance the writing, potentially by focusing solely on presenting pivotal and insightful findings in the main paper."

---

### Decision · Program_Chairs · 2023-09-21

**Decision:**

Accept (poster)

**Comment:**

The paper provides a theoretical study of reinforcement learning from preferences. The paper provides a reduction of preference based RL to other RL settings and shows that in many cases the additional cost incurred by those reductions are small. For preferences that can be explained by an underlying unknown reward function, then the authors provide a reduction to reward-based RL. For more general preferences, the paper provides a reduction to finding a Nash-equilibrium in 2-player factored Markov games.

There was consensus among the reviewers that the paper provides a solid contribution to literature. While the results are not particularly surprising, the analysis is sound and well executed. There were initial concerns that the results are only applicable to simple / small MDPs but the authors could clarify this point in their rebuttal and thus, these concerns may instead show an issue of presentation. In fact, clarity and presentation was assessed very differently across reviewers. While some sound the paper well written and clear, others found it hard to follow. The AC found the paper to be clear but believes that this divide between reviewers' assessment on the presentation is due to 2 things:

1. The paper requires a certain background and whether the paper is easy to follow heavily depends on whether a reader is already very familiar with specific concepts such as Bellman eluder dimension or von Neumann winner etc. It would be advisable for the authors to provide more background in order to make the paper accessible to a wider audience, perhaps with examples and further pointers to the literature.
2. The title (and introduction) are very broad, which may raise expectations on the paper that are then not met. Given that RLHF is currently very popular for fine-tuning complex neural network models, readers may indeed be disappointed to see a rather abstract treatment that is somewhat removed from neural network models. Of course, the reductions are agnostic to the function approximator used (and the authors give explicit examples of problems with small BE-dim) but it is also not clear that the results indeed provide direct implications for RLHF fine tuning of neural network models. To avoid this mismatch in expectations, the authors are advised to emphasize in the title and introduction the nature of the contributions. Perhaps highlight the theoretical focus by appending "-- A theoretical study" to the title?

Overall, no consensus could be reached in the reviewer discussion. Based on the ACs own reading of the paper, the AC agrees that the paper does provide valuable contributions to the theory of preference-based RL, and believes that the weaknesses stem from issues in the presentation that can be addressed with smaller modifications in a camera ready version. As a result, acceptance is recommended.